# Complex nitrogen sources from agro-industrial byproducts: impact on production, multi-stress tolerance, virulence, and quality of *Beauveria bassiana* blastospores

Valesca Henrique Lima,[1,2] Alexandre Toshihiro Matugawa,[2] Gabriel Moura Mascarin,[3] Éverton Kort Kamp Fernandes[1,2]

**ABSTRACT** We investigated the impact of various complex organic nitrogen sources on the submerged liquid fermentation of *Beauveria bassiana*, a versatile entomopathogenic fungus known for producing hydrophilic yeast-like single cells called blastospores. Specifically, we examined yeast extract, autolyzed yeast, inactive yeast, cottonseed flour, corn bran, and corn gluten meal as nitrogen compounds with different carbon-to-nitrogen (C:N) ratios. Our comprehensive analysis encompassed blastospore production, tolerance to abiotic stresses, shelf stability after drying, and virulence against mealworm larvae, crucial attributes for developing effective blastospore-based biopesticides. Notably, cottonseed flour emerged as the optimal nitrogen source, yielding up to $2.5 \times 10^9$ blastospores/mL within 3 days in a bioreactor. These blastospores exhibited the highest tolerance to heat stress and UV-B radiation exposure. The endogenous C:N ratio in blastospore composition was also impacted by nitrogen sources. Bioassays with mealworm larvae demonstrated that blastospores from cottonseed flour were the most virulent, achieving faster lethality (lower $LT_{50}$) and requiring a lower inoculum ($LC_{50}$). Importantly, blastospores produced with cottonseed flour displayed extended viability during storage, surpassing the retention of viability compared to those from autolyzed yeast over 180 days at 4°C. Despite differences in storage viability, both nitrogen sources conferred similar long-term blastospore bioactivity against mealworms. In summary, this research advances our understanding of the crucial impact of complex organic nitrogen selection on the phenotypic traits of blastospores in association with their intracellular C:N ratio, contributing to the production of ecologically fit, shelf-stable, and virulent propagules for effective pest biocontrol programs.

**IMPORTANCE** Biological control through entomopathogenic fungi provides essential ecological services in the integrated management of agricultural pests. In the context of submerged liquid fermentation, the nutritional composition significantly influences the ecological fitness, virulence and quality of these fungi. This study specifically explores the impact of various complex organic nitrogen sources derived from agro-industrial byproducts on the submerged liquid fermentation of *Beauveria bassiana*, a versatile entomopathogenic fungus known for producing hydrophilic yeast-like blastospores. Notably, manipulating the nitrogen source during submerged cultivation can influence the quality, fitness, and performance of blastospores. This research identifies cottonseed flour as the optimal low-cost nitrogen source, contributing to increased production yields, enhanced multi-stress tolerance, heightened virulence with extended shelf life and long-term bioactivity. These findings deepen our understanding of the critical role of nitrogen compound selection in liquid media formulation, facilitating the production of ecologically fit and virulent blastospores for more effective pest biocontrol programs.

Address correspondence to Éverton Kort Kamp Fernandes, evertonkort@ufg.br, or Gabriel Moura Mascarin, gabriel.mascarin@embrapa.br.

The authors declare no conflict of interest.

See the funding table on p. 22.

**KEYWORDS** biocontrol, entomopathogenic fungi, multivariate analysis, phenotypic traits, thermotolerance, UV-B radiation

In a world marked by the emergence of pesticide resistance and human and environmental health concerns, several studies on natural products have been developed as non-chemical strategies for sustainable pest control (1, 2). Mycoinsecticides and mycoacaricides, composed of fungal propagules from various entomopathogenic species, represent significant alternatives to replace or complement conventional synthetic chemical pesticides (3). Recognized as eco-friendly tools, they play a vital role in integrated pest management programs worldwide, contributing to the commercialization of at least 170 fungal products since the 1960s (4).

One of the most commercialized biopesticides worldwide is the cosmopolitan entomopathogenic fungus *Beauveria bassiana* (Bals.) Vuill. (Ascomycota: Cordycipitaceae). This fungus plays a relevant role in the biological control of numerous arthropod pests of medical, veterinary, and agricultural importance. Furthermore, this fungus can facultatively form intimate associations with plants as endophytes, resulting in improved plant nutrition and resilience to biotic and abiotic stresses (5–8). Mass production technologies are imperative for commercializing this fungus to achieve large amounts of propagules required for its widespread applications in various agricultural settings and global commercialization (9). Aerial conidia are the primary infective fungal propagules and are easily obtained by solid substrate fermentation using typically pre-cooked cereal grains. This type of propagule stands out as the most investigated fungal active ingredient for use in biological control strategies designed to control arthropod pests and, hence, the most commercially exploited by industry. However, a different form of propagule, known as blastospore, is a vegetative, yeast-like, single cell that originates from the dimorphic growth of hypocrealean entomopathogenic fungi during the infection stage inside the host hemocoel. This transition involves the fungus switching from filamentous growth (hyphae) to yeast-like growth (unicellular blastospores). The *in vitro* production of blastospores is accomplished through submerged liquid fermentation technology, closely simulating the nutrient-rich hemolymph of insects and promoting the prolific growth of these yeast-like cells. This fermentation technology is considered more advantageous than solid substrate fermentation owing to its ease of scalability and downstream processing, higher yields achieved in shorter cultivation periods, and substantial reductions in operation costs, labor, and physical space. Moreover, it is less prone to contamination due to more rigorous control of the fermentation parameters and aseptic conditions (10, 11). Despite their physical distinctions, both conidia and blastospores are susceptible to abiotic factors like heat and UV-B stresses. Consequently, ensuring their efficacy against arthropod pests demands proper handling, production, and formulation techniques (12–15).

The process of producing fungi can be manipulated to obtain high concentrations of stress-tolerant propagules, resulting in more efficient and ecologically sound bioproducts (16–21) produced with the cheapest ingredients possible (22). The type of fermentation can positively impact the production of blastospores, their ability to resist dehydration, and their storage stability. Nitrogen is an important nutrient for the growth and development of fungi (7, 11, 23, 24) but is costly in liquid media (11). Manipulation of nitrogen sources in liquid media provides a means to enhance production yields (11, 18, 23, 25–27), virulence to arthropods (26–28), desiccation tolerance (11, 18, 23, 25, 29, 30), and shelf life of blastospores of entomopathogenic fungi (11, 29). Previous studies have shown that the levels of lipids, glycogen, and protein in dry biomass of blastospores, as well as the presence of intracellular trehalose and mannitol, are crucial for their ability to withstand desiccation and heat stresses. These cellular compounds are affected by factors such as the C:N ratio and carbon and nitrogen sources used as nutrients in the culture media (23, 25). Although *B. bassiana* development demands organic nitrogen sources, complex nitrogen compounds appear to be particularly effective in producing high amounts of desiccation-tolerant blastospores (11, 20, 23,

26, 29). The utilization of complex organic substrates for growing *B. bassiana* blastospores through submerged liquid fermentation in benchtop stirred-tank bioreactors, along with subsequent assessment of their insecticidal effectiveness, desiccation and UV-B tolerance, storage stability, and bioactivity persistence, has not been systematically examined. Furthermore, the quality and fitness of blastospores produced through this process have not been thoroughly investigated and correlated with compositional analyses of their endogenous cellular C:N ratio.

This study aims to comprehensively examine the influence of diverse nitrogen sources derived from complex organic proteins available as agro-industrial byproducts on the production yield, quality, ecological fitness, shelf stability, and biocontrol effectiveness of *B. bassiana* blastospores. To this end, we investigated six different proteinaceous substrates to compare the multiplication, thermotolerance, UV-B tolerance, and virulence of blastospores produced through submerged liquid fermentation. The endogenous C:N ratio of the resulting blastospores was determined based on their dry matter. Furthermore, we explored the impact of the two most effective nitrogen sources on the prolonged storage stability of blastospores formulated with diatomaceous earth, followed by the assessment of their bioactivity persistence against insects.

Our study advocates for an integrative approach, relying on multiple phenotypic traits, to guide the selection of appropriate nitrogen compounds for the enhanced mass production of *B. bassiana* blastospores through submerged liquid culture. Building upon our prior research (11), it further emphasizes the significance of carefully selecting agro-industrial nitrogen substrates to improve various biological and ecological traits of blastospores. This approach seeks to uphold the tolerance of blastospores to abiotic stresses while simultaneously enhancing the virulence of these propagules for effective control of arthropod pests, including insects, mites, and ticks.

## MATERIALS AND METHODS

### Fungal strain and inoculum preparation

*Beauveria bassiana* strain IP 361 was initially isolated in 2010 from *Amblyomma sculptum* (Acari, Ixodidae) adults in Central Brazil; this fungal strain is a good blastospore producer (>$10^8$ blastospores/mL in liquid medium) and notably virulent against the cattle tick *Rhipicephalus microplus* (31). The fungus was cultured in Petri plates (90 × 15 mm) on potato dextrose agar medium (Difco Laboratories, Sparks, MD, USA) supplemented with 0.1% yeast extract (Difco Laboratories, Interlab, São Paulo, SP, Brazil) (PDAY) at 27°C ± 1°C, relative humidity (RH) > 80%, and 12:12 h light:dark (L:D) photoperiod for 15 days.

The inoculum was prepared with fresh conidia harvested from the agar cultures using a spatula and suspended in 10 mL Tween 80 0.01% (vol/vol). Conidial suspensions were vortexed and filtered through a sterile cheesecloth (pore size ~ 30 µm). Conidia were quantified in a hemocytometer at 400× magnification under a light microscope (Nikon Eclipse E200, Nikon Instruments, Inc.), and subsequently, the concentrations were adjusted to 1 × $10^8$ conidia/mL.

Pre-cultured blastospores were obtained in an Adamek-modified liquid medium (31–33). Briefly, for each liter of distilled water, we added 3% starch broth [0.6 g corn flour (Maizena, São Paulo, SP, Brazil) in 30 mL distilled water], 4% (wt/vol) yeast extract (Difco Laboratories, Interlab, São Paulo, SP, Brazil), 4% (wt/vol) glucose (Labsynth Products Laboratories Ltda., Diadema, SP, Brazil), and 0.4% (vol/vol) Tween 80 (Labsynth Products Laboratories Ltda., Diadema, SP, Brazil). Pre-cultures were grown in 100 mL medium with 250-mL Erlenmeyer flasks sealed with a hydrophobic cotton plug to allow gas exchange and aeration. Each culture flask was then inoculated with 1 mL of a conidial suspension to deliver a final concentration of 1 × $10^6$ conidia/mL. The flasks were incubated in an orbital shaker (TE-422, Tecnal, Piracicaba, SP, Brazil) at 27°C ± 1°C, 12:12 h (L:D) and 180 rpm for 4 days.

**TABLE 1** Total organic carbon and nitrogen contents, carbon-to-nitrogen ratio, and cost of different complex organic nitrogen sources used to produce *Beauveria bassiana* blastospores in submerged liquid fermentation

| Nitrogen source | Carbon (%) | Nitrogen (%) | C:N ratio[a] | Cost per L (USD)[b] |
|---|---|---|---|---|
| Yeast extract | 40.0 | 10.9 | 3.67:1 | 52.97 |
| Autolyzed yeast | 47.1 | 7.02 | 6.71:1 | 0.35 |
| Inactive yeast | 47.9 | 6.96 | 6.88:1 | 0.27 |
| Cottonseed flour | 49.3 | 9.92 | 4.97:1 | 0.89 |
| Corn bran | 47.3 | 3.87 | 12.22:1 | 0.06 |
| Corn gluten meal | 53.9 | 11.55 | 4.67:1 | 0.27 |

[a]Carbon-to-nitrogen ratio.
[b]Considering 1.0 USD = 4.95 BRL on 29 January 2024, all nitrogen sources were employed at 30 g/L of liquid medium. Glucose, minerals, and vitamins are all kept constant in all liquid media, regardless of the nitrogen source.

## Blastospore cultivation in liquid media with different nitrogen sources

Six organic nitrogen sources were chosen to supplement the liquid medium where *B. bassiana* blastospores were grown: (i) yeast extract (control; Difco Laboratories, Interlab, São Paulo, SP, Brazil); (ii) autolyzed yeast (Lyscell, ICC, São Paulo, SP, Brazil); (iii) inactive yeast (StarYeast, ICC, São Paulo, SP, Brazil); (iv) cottonseed flour (Pharmamedia, ADM Co., Decatur, IL, USA); (v) corn bran (Refinazil, Ingredion, Mogi Guaçu, SP, Brazil); and (vi) corn gluten meal (Protenose, Ingredion, Mogi Guaçu, SP, Brazil). The total carbon and nitrogen contents, followed by the C:N ratio of each substrate, are shown in Table 1.

The final liquid medium consisted of 3% (wt/vol) nitrogen source, 12% (wt/vol) glucose, and 50% (vol/vol) basal salts supplemented with trace metals and vitamins as follows (amount per liter): $KH_2PO_4$, 2.0 g; $CaCl_2 \cdot 2H_2O$, 0.4 g; $MgSO_4 \cdot 7H_2O$, 0.3 g; $CoCl_2 \cdot 6H_2O$, 37 mg; $FeSO_4 \cdot 7H_2O$, 50 mg; $MnSO_4 \cdot H_2O$, 16 mg; $ZnSO_4 \cdot 7H_2O$, 14 mg; thiamin, riboflavin, pantothenate, niacin, pyridoxamine, and thioctic acid, 500 mg each; folic acid, biotin, and vitamin B12, 50 mg each (11, 16). The medium without glucose and vitamins was first autoclaved for 20 min at 121°C. Glucose stock solution at 25% (wt/vol) was autoclaved separately at the same conditions. The vitamin solution was filtered through a 0.22-µm-pore size Millipore membrane (Merck, Darmstadt, Germany) and added to the autoclaved medium prior to inoculation.

After preparation, 50 mL of each medium was kept in 250 mL Erlenmeyer flasks, inoculated with the pre-cultured blastospores (2% vol/vol inoculum) to deliver the final concentration of $5 \times 10^6$ blastospores/mL, and sealed with a hydrophobic cotton plug. The culture media were incubated in an orbital shaker at 27°C ± 1°C and 350 rpm for 3 days. The flasks were shaken manually once a day to minimize the mycelial growth on the flask walls. After fungal growth, the culture media were filtered through cheesecloth (~30 µm pore mesh) to remove the mycelia and collect only filtered blastospores with 50-mL centrifuged tubes. The blastospore suspensions were centrifuged at 4,200 rpm for 5 min at 16°C, the supernatant discarded, and the pellets resuspended in Tween 80 0.01% (vol/vol). This procedure was repeated twice to remove any residual spent medium. Blastospores from each liquid culture were quantified using a hemocytometer at 72-h post-inoculation under a phase-contrast microscope with 400× magnification. Two counts (*i.e.*, technical replicates) were performed for each flask, with two biological replicates (flasks) for each medium. The entire experiment was repeated at least three times on different occasions using fresh fungal inoculum.

## Assays for assessing blastospores' tolerance to heat or UV-B radiation

Blastospores produced from each medium supplemented with a particular nitrogen source were suspended in Tween 80 0.01% (vol/vol) as mentioned above, and the concentration was adjusted to $1 \times 10^3$ blastospores/mL. Two milliliters of each blasto-spores suspension were transferred to 16 × 125 mm glass screw-cap tubes (Pyrex, Corning, São Paulo, SP, Brazil) and immediately heated in a water bath at 45°C ± 0.2°C. The suspensions were exposed to heat for 0 (control), 15, 30, 45, 60, 90, 120, 150, or

180 min. Temperature was monitored with a HOBO H8 data logger (Onset Computer Corporation, Bourne, MA, USA) throughout the heat exposure. Then, an aliquot of 50 µL of each sample was spread using a Drigalski spatula on PDAY medium in Petri plates (90 × 15 mm), supplemented with 0.05% (wt/vol) chloramphenicol.

UV-B tolerance tests were performed by spreading 50 µL of blastospores suspension ($1 \times 10^3$ blastospores/mL) on PDAY medium supplemented with 0.05% (wt/vol) chloramphenicol in Petri plates (90 × 15 mm), using a Drigalski spatula. The plates were immediately covered with a 0.13 mm thick cellulose diacetate film (JCS Industries, Le Mirada, CA, USA), which blocks UV-C radiation (below 280 nm) and the UV-B short-wavelength (280–290 nm) but permits the passage of most UV-B (290–320 nm) and the minimal UV-A (320–400 nm) radiation. The plates were then exposed to 743.75 mW/m$^2$ of Quaite-weighted irradiance (34, 35) for 0 (control), 30, 60, 90, 120, 150, or 180 min in a chamber containing four ultraviolet lamps (UVB-313 EL/40W; Q-Lab Corporation, Westlake, OH, USA), which corresponded to fluences of 0 (control), 1.4, 2.7, 4.1, 5.4, 6.8, and 8.1 kJ/m$^2$ of UV-B radiation, respectively (36, 37). The spectral irradiance was measured with a USB 2000+ Rad Spectroradiometer (Ocean Optics, Dunedin, FL, USA). The temperature inside the irradiation chamber was monitored with a HOBO data logger.

After exposure to heat or UV-B radiation, the culture plates were incubated for 7 days at 27°C ± 1°C, RH > 80%, and 12:12 h L:D photoperiod. The susceptibility to heat and UV-B stresses was assessed by counting the number of colonies (colony-forming units—CFU) in each plate and calculating the normalized relative viability (RV) of blastospores in each exposure time in relation to the control group (untreated; time point of 0), as follows: RV = (number of colonies in the group exposed to heat or UV-B/number of colonies in the control group) × 100 (36).

## Insect bioassays for assessing fungal virulence

Blastospores obtained from culture media with different nitrogen sources were assayed against larvae (18–20 mm in length) of *Tenebrio molitor* (Coleoptera: Tenebrionidae). These mealworm larvae were obtained from an insect colony maintained in the Instituto de Patologia Tropical e Saúde Pública at Universidade Federal de Goiás, Goiás, Brazil. Previously to bioassays, the insects were immersed in 0.1% sodium hypochlorite solution for 15 s, rinsed in distilled water, and dried with sterile tissue paper. This procedure was repeated twice.

The virulence of blastospores was tested at four concentrations ($1 \times 10^5$, $1 \times 10^6$, $1 \times 10^7$, and $1 \times 10^8$ blastospores/mL). Experimental units consisted of 15 larvae immersed for 30 s in 20 mL of blastospore suspension or 20 mL of sterile 0.01% Tween 80 solution (control). Mealworms were subsequently transferred into Petri plates covered with cellulose filter papers (Qualy, Ø 12.5 cm, 80 g/m$^2$) and allowed to air dry for 15 min. Fresh pieces of organic cabbage were provided as food and water sources. Treated mealworms were incubated at 27°C ± 1°C, RH > 90%, and 12:12 h L:D photoperiod for 15 days. Mortality was recorded daily, and dead larvae were transferred to moist chambers to confirm mycosis by external sporulation on the cadavers (38).

## Determination of endogenous carbon-to-nitrogen ratio in dry matter of blastospores

To assess the impact of the different nitrogen sources on the quality of blastospores, we determined the endogenous carbon and nitrogen composition and their C:N ratio from dried blastospore biomass previously harvested from 3-day-old liquid cultures cultivated under the same conditions described above. Blastospores were harvested and filtered through a nylon mesh (40 µm pore size) to remove mycelia and obtain clean blastospore suspensions. The suspensions were centrifuged for 10 min at 5,000 rpm and 4°C, and the supernatants were discarded. The resulting biomass was resuspended in PBS buffer (pH 7.0), centrifuged at the same conditions above, and the supernatant was discarded to remove all media components. This procedure was repeated twice. Subsequently, the blastospore pellets were freeze-dried in a benchtop freeze dryer (L101, Liotop, São

Carlos, SP, Brazil) at −45°C for 12 h and macerated to obtain a fine powder. Three independent biological replicates for blastospores produced in culture media with each nitrogen source were analyzed.

The endogenous carbon and nitrogen contents from the blastospore samples on a dry-mass basis were quantified through a CHNS analysis, a modified Dumas method. Approximately 2–3 mg of lyophilized blastospores was loaded into tin capsules and combusted at 900°C under a continuous flow of helium and oxygen in a Thermo Scientific FLASH 2000 Organic Elemental Analyzer. The detection system employed is a thermal conductivity detector, and the calibration curve was constructed using sulfanilamide as the standard (P/N 338 25100).

## Kinetic analysis of blastospore production and storage studies

The kinetics of *B. bassiana* IP 361 blastospores scaled-up production in a culture medium supplemented with autolyzed yeast or cottonseed flour as an organic nitrogen source was investigated. In addition, blastospores were formulated, and their shelf life was assessed. *B. bassiana* was pre-cultured in liquid Adamek-modified medium as described in section "Fungal strain and inoculum preparation," and the scaled-up production of blastopores was conducted in a 1 L bioreactor (TEC-BIO-1.5, Tecnal, Piracicaba, SP, Brazil), with 650 mL of each supplemented culture medium. The liquid culture media were inoculated with blastospores from the pre-cultures to obtain the initial concentration of $5.0 \times 10^6$ blastospores/mL. The medium was agitated in a bioreactor at 400 rpm, 27°C ± 1°C, and an aeration rate of 1.5 L/min for 72 h. Three experiments were carried out on different days to obtain three batches of blastospores.

The kinetics of blastospores production was assessed by collecting 25 mL of destructive samples every 12 h for 72 h of cultivation. Each sample was assessed for the number of blastospores, the pH of the culture medium, and the glucose consumption. Blastospores were quantified in a hemocytometer, the pH was measured using a pH meter (K39-2014B, Kasvi, São José dos Pinhais, PR, Brazil), and the amount of glucose was determined by the quantification of carbohydrates using the 3,5-dinitrosalicylic acid method (39).

Seventy-two hours after cultivation, the blastospores were harvested, quantified, and separated from the spent medium by adding 1.0 g of autoclaved diatomaceous earth (DE) for every $2 \times 10^{10}$ blastospores (9). The mixture of blastospores and DE was filtered with a vacuum pump, using a qualitative filter paper (Unifil, Ø 12.5 cm, 80 g/m$^2$) coupled to a Büchner funnel and a Kitasato flask. The resulting cake was divided into smaller portions, dried in a laminar flow hood between 20 and 24 h, ground with a mortar and pestle, and sieved (18 mesh, 1.0 mm). The moisture content was evaluated using an infrared balance (LJ16 Moisture Analyzer, Mettler Toledo, Laboratory & Weighing Technologies, Switzerland), and only samples with a maximum of 6% moisture were tested. Air-dried blastospores were stored in sterile 50 mL centrifuge tubes (K19-0050, Kasvi, São José dos Pinhais, PR, Brazil) at 4°C (refrigerator) to perform shelf-life evaluations.

The stability of air-dried blastospores was evaluated, considering their viability and virulence against *T. molitor* larvae. The time points measured were 1 day (blastospores newly produced and immediately dried), 1 week, 2 weeks, 1 month, 2 months, 3 months, and 6 months of storage. Each sample of 0.5 g dried blastospores was suspended in 10 mL of 0.01% Tween 80, then the blastospores were quantified in a hemocytometer, and the suspension was adjusted to $1.0 \times 10^6$/mL. An aliquot of 30 µL of each suspension was inoculated on PDAY medium supplemented with 0.05% (wt/vol) chloramphenicol and 0.002% (wt/vol) benomyl (50% active ingredient; Benlate, DuPont, São Paulo, SP, Brazil) in Petri plates (35 × 10 mm). The plates were incubated at 27°C ± 1°C and 12:12 h L:D photoperiod for 24 h; after this period, a drop of lactophenol cotton blue stain was placed on the inoculum and allowed to dry. Using a phase-contrast microscope at 400× magnification, the germination of a minimum of 300 blastospores per plate was evaluated and counted in triplicate. Blastospores with germ tubes larger than the

cell size were deemed germinated (viable). The relative germination was calculated according to Braga et al. (36) through the equation: relative germination (%) = (Gt/Gc) × 100, where Gt is the number of blastospores germinated at each evaluated time, and Gc corresponds to the average number of blastospores germinated in the control group. The virulence of stored blastospores was also assayed against *T. molitor* larvae as a means to measure the bioactivity persistence of these propagules. Experimental units consisted of 20 larvae (18–20 mm in length) immersed for 30 s in 10 mL of blastospores' suspension at $1.0 \times 10^7$ blastospores/mL following the method described in section "Insect bioassays for assessing fungal virulence". The evaluation time points were the same as those chosen to investigate the viability of blastospores in PDAY during cold storage.

## Data analysis

The production of blastospores was fit to a linear model with normal distribution with nitrogen source being the only predictor, and significance was assessed through one-way analysis of variance (ANOVA). A quadratic regression model was fitted to blastospore yields obtained with cottonseed flour and autolyzed yeast in the function of the fermentation time. Regression curves were compared with the *F*-test. Relative viability data for tolerance of blastospores to UV-B radiation or heat stress were fitted to a two-parameter Weibull model using the package "drc" (40). Median lethal times ($LT_{50}$) were estimated after fitting a parametric Weibull model with binomial distribution to these data sets using the package "survival" (41), and these estimates were compared with the Tukey HSD test at $P < 0.05$. Virulence was measured as a means of $LT_{50}$ across fungal concentrations and median lethal concentrations ($LC_{50}$), after fitting mortality data recorded at days 5 and 7 post-inoculation to a two-parameter Gompertz model with binomial distribution using the package "drc." Shelf life data expressed in CFU were analyzed by fitting the relative viability measurements to a generalized linear mixed model with binomial to compare the effect of two nitrogen sources on the blastospore shelf stability during 180 days of cold storage. The bioactivity of air-dried blastospores during storage at 4°C was determined using survival analysis and comparing the survival curves across storage time points with a log likelihood ratio test (LRT) ($P < 0.05$). The endogenous C:N ratio in blastospore composition (on a dry matter basis) was subjected to a linear model with normal distribution and then one-way ANOVA, followed by comparison of means based on Tukey HSD test ($P < 0.05$). The principal component analysis (PCA) was used to assess the relationships between phenotypic variables and nitrogen sources tested and associate the correlated parameters using the package "FactoMineR" (42). All statistical analyses and were performed with an R statistical environment (43).

## RESULTS

### Nitrogen compounds affect blastospore production yields

The mean production of blastospores of *B. bassiana* s.l. IP 361 cultured in liquid media with different nitrogen sources for 72 h varied significantly ($F = 5.21$, df = 5, 30, and $P = 0.0014$). The culture media supplemented with cottonseed flour or autolyzed yeast had a pronounced blastospore production, with $1.52 \times 10^9$ and $1.45 \times 10^9$ blastospores/mL, respectively, which was significantly higher than the production reached in the culture media supplemented with yeast extract or corn bran ($7.67 \times 10^8$ and $8.12 \times 10^8$ blastospores/mL) (Fig. 1).

### Nitrogen compounds affect blastospore thermotolerance

The response of *B. bassiana* blastospores to wet heat (45°C) was revealed by survival rates that decreased significantly as the exposure time increased (Fig. 2A). The nitrogen sources used in the fermentation media produced blastospores with phenotypes exhibiting varied degrees of susceptibility to heat stress as noted by remarkable

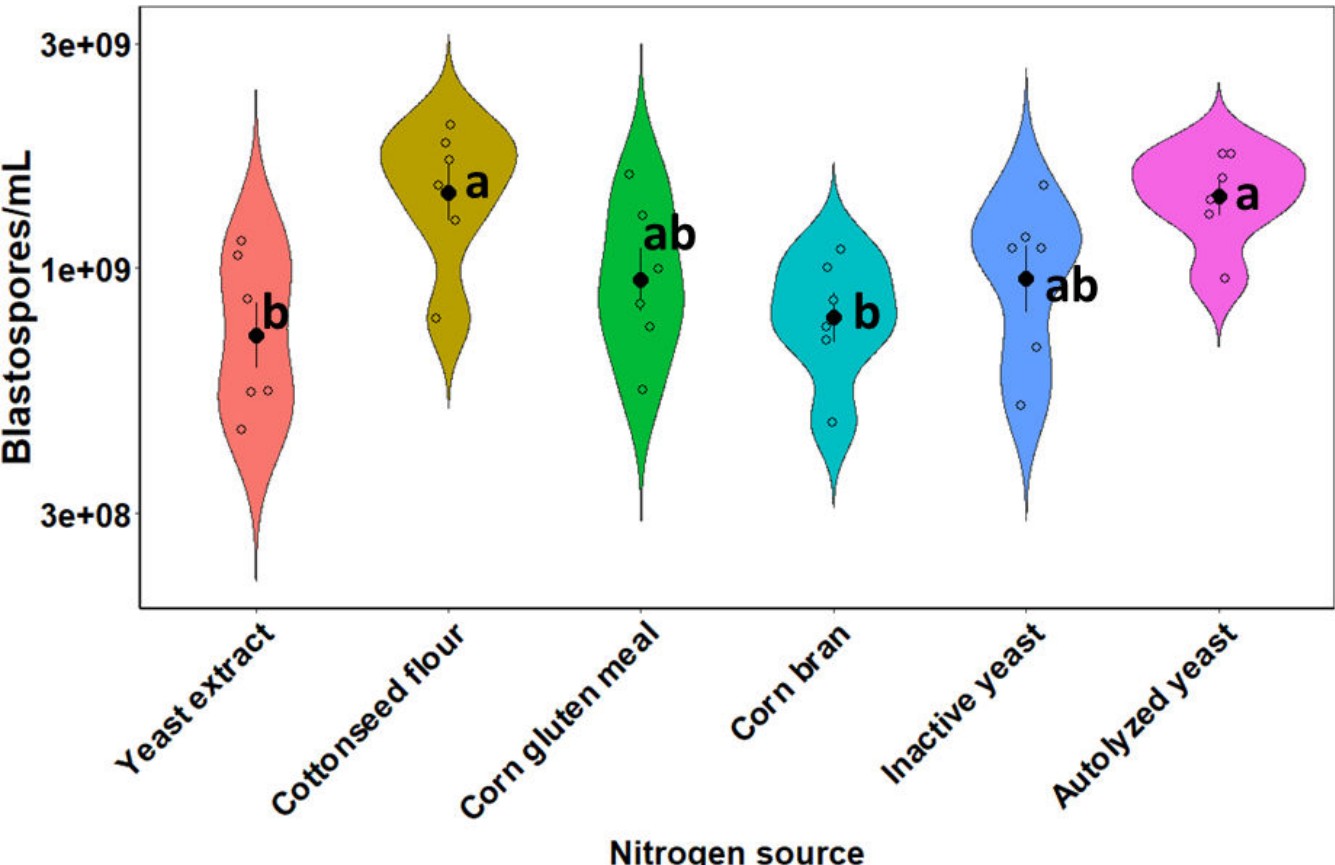

**FIG 1** Violin plots show blastospore yield of *Beauveria bassiana* s.l. IP 361 grown in liquid cultures with different nitrogen sources and incubated at 27°C and 350 rpm for 72 h. Black dot and error bar inside each violin plot represent the mean and standard error (±SE), respectively. Empty dots represent observational data (*n* = 6 replicates). Different letters by the box plots indicate statistically significant differences in blastospore yield between treatment groups at *P* < 0.05.

differences among their survival rates (LRT = 62.452, df = 15, *P* < 0.0001) (Fig. 2B). Blastospores derived from cottonseed flour stood out as the most tolerant to heat stress. Their mean relative survival rate was ~45% longer (53.99 min) than those blastospores produced in culture medium supplemented with corn bran, which attained the shortest survival time (29.89 min) (Fig. 2B). Overall, yeast extract, inactive yeast, and cottonseed flour supplemented in the culture media for blastospore growth attained the most extended survival rates under heat stress. Blastospores did not survive after 180 min of heat exposure (0% viability), regardless of the nitrogen source tested.

### Nitrogen compounds had no effect on blastospore UV-B tolerance

Blastospores subjected to UV-B radiation displayed a notable decline in survival trends, showing a significant decrease as the UV-B dose increased, ranging from 0 to 8.1 kJ/m$^2$ (Fig. 3A). No effect was observed due to nitrogen sources on the UV-B tolerance of blastospores (LRT = 13.398, df = 15, and *P* = 0.572) (Fig. 3B). Notably, lethal UV-B fluences that inactivated 50% of blastospores produced by submerged cultivation with different nitrogen sources varied from 3.57 to 4.70 kJ/m$^2$. Despite the nitrogen source tested, blastospores were entirely inactivated after exposure to UV-B at 8.1 kJ/m$^2$.

### Nitrogen compounds affect blastospore virulence

The insect bioassays demonstrated that the nitrogen sources used to supplement the fermentation culture medium significantly affected the speed of blastospores to kill the mealworm larvae within each fungal concentration tested (Fig. 4A). However, mealworm

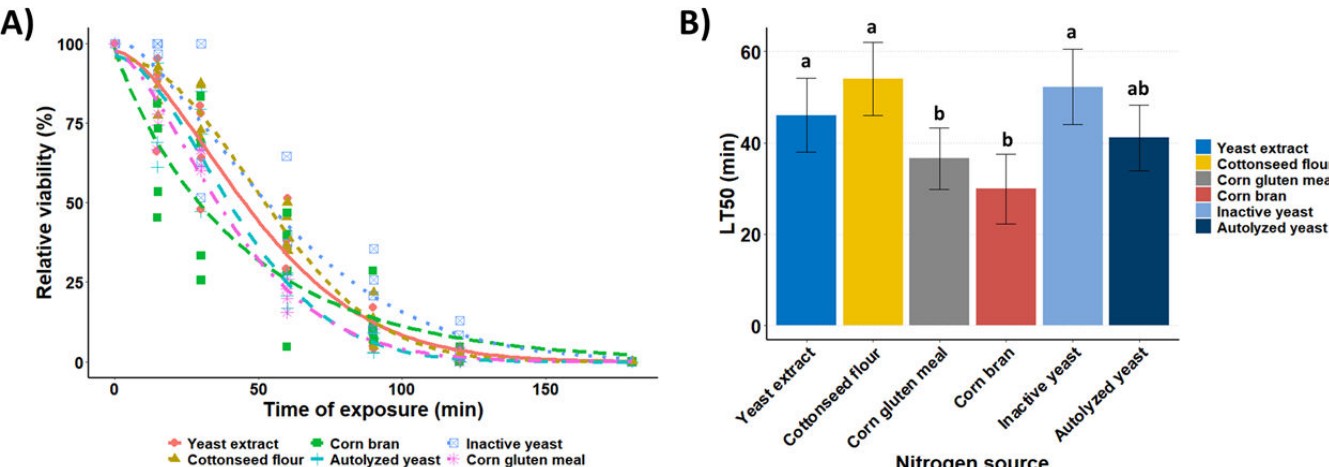

**FIG 2** Impact of nitrogen sources on the relative viability of *Beauveria bassiana* s.l. IP 361 blastospores after exposure to crescent intervals of heat shock (45°C ± 0.2°C). (A) Three-parameter Weibull model curves fitted to relative blastospore viability based on CFU calculated in relation to unheated controls, while symbols represent original data points. (B) Estimated median lethal times ($LT_{50}$, minutes), followed by their respective 95% confidence intervals of blastospores produced with different nitrogen sources after heat exposure. A representative experiment of six independent replications, where $LT_{50}$ values followed by the same letters indicate non-statistical differences ($P < 0.05$).

mortality significantly increased as time progressed, irrespective of nitrogen sources and inoculum concentrations tested. All time-mortality trends fit the parametric Weibull model well, irrespective of nitrogen sources and the concentration of blastospores. All survival trends of mealworms were clearly reduced when treated with blastospores derived from culture media enriched with different nitrogen sources compared to the untreated control (LRT = 1,426, df = 9, and $P < 0.0001$). Corn bran, inactive yeast, and cottonseed flour consistently yielded blastospores that demonstrated heightened virulence toward mealworms, particularly at lower concentrations tested ($1 \times 10^5$ and $1 \times 10^6$). They delivered shorter lethal times (~47%–54% lower $LT_{50}$) than the blastospores produced in liquid media supplemented with yeast extract, corn gluten meal, or autolyzed yeast (Fig. 4B). The fastest mortality rates, as seen by the sharp decrease in the

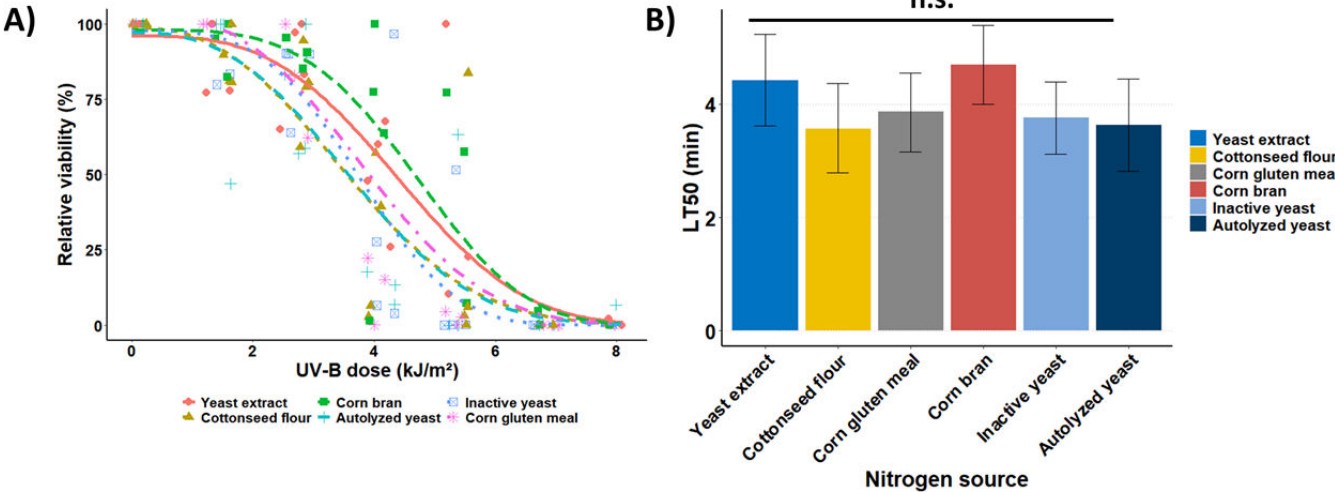

**FIG 3** Impact of nitrogen sources on the relative viability of *Beauveria bassiana* s.l. IP 361 blastospores after exposure to UV-B Quaite-weighted irradiance (743.75 mW/m²) of 0 (control), 1.4, 2.7, 4.1, 5.4, 6.8, or 8.1 kJ/m². The relative viability of CFU was calculated in relation to non-irradiated controls. (A) Weibull model curves fitted to relative blastospores' viability, while symbols represent original data points. (B) Estimated median lethal doses ($LD_{50}$, kJ/m²) followed by their respective 95% confidence intervals of blastospores produced with different nitrogen sources after UV-B exposure. A representative experiment of six independent replications, where $LD_{50}$ values were not statistically different (n.s.) among treatment groups ($P < 0.05$).

## A) Mealworm survival trends

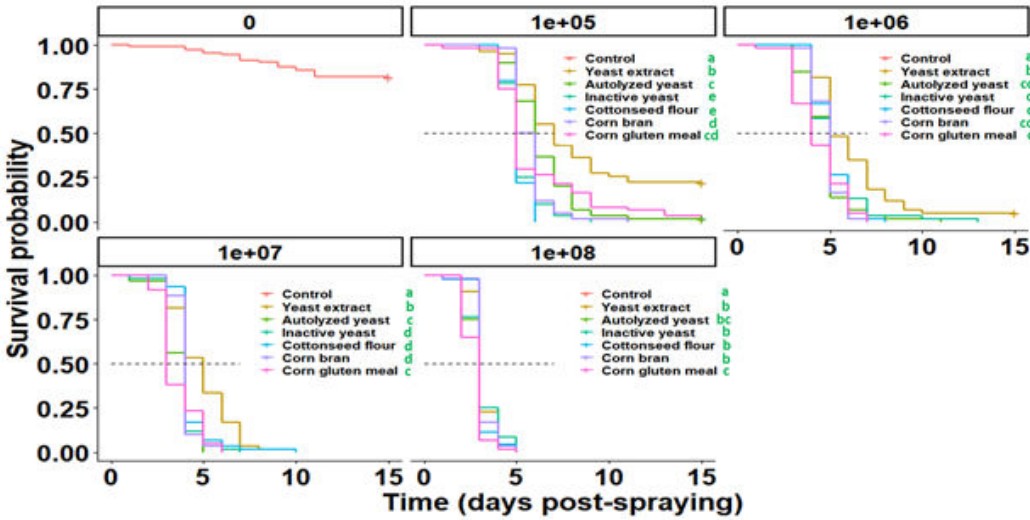

## B) Estimated median lethal time

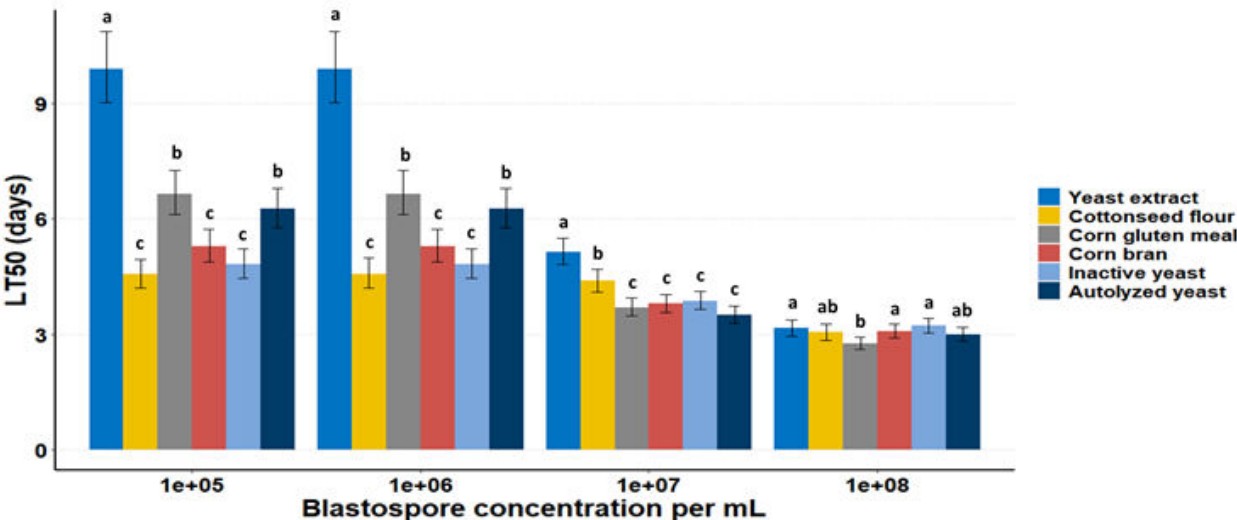

FIG 4 Survival of *Tenebrio molitor* larvae treated with *Beauveria bassiana* s.l. IP 361 blastospores produced from liquid fermentation with different complex nitrogen sources. (A) Mean survival (%) and 95% confidence intervals of mealworm larvae inoculated with *B. bassiana* blastospores, and non-significant differences between survival curves are indicated by the same letters according to the log-rank test ($P < 0.05$). (B) LT$_{50}$ values of *Tenebrio molitor* larvae exposed to four application rates of *Beauveria bassiana* blastospores estimated using the parametric Weibull model. Means (±95% confidence intervals) followed by the same letters indicate non-statistical differences ($P < 0.05$). A representative experiment of four independent replications.

insect survival trends, were achieved at $1 \times 10^8$ blastospores/mL, wherein 100% mortality was reached within 5 days post-infection, irrespective of nitrogen sources used to produce blastospores. All dead insects exhibited mycosis and then were confirmed to be infected by *B. bassiana*.

The effect of inoculum concentration on the insecticidal activity of blastospores against mealworm larvae was recorded in two post-infection time points (day 5: $\chi^2 =$ 107.359, df = 15, and $P < 0.0001$; day 7: $\chi^2 = 170.52$, df = 10, and $P < 0.0001$) (Fig. 5A). At days 5 and 7 post-infection, mealworm mortality significantly increased with the increase of blastospores concentration, irrespective of nitrogen sources used for their production. All concentration-mortality trends fit well with the two-parameter Gompertz

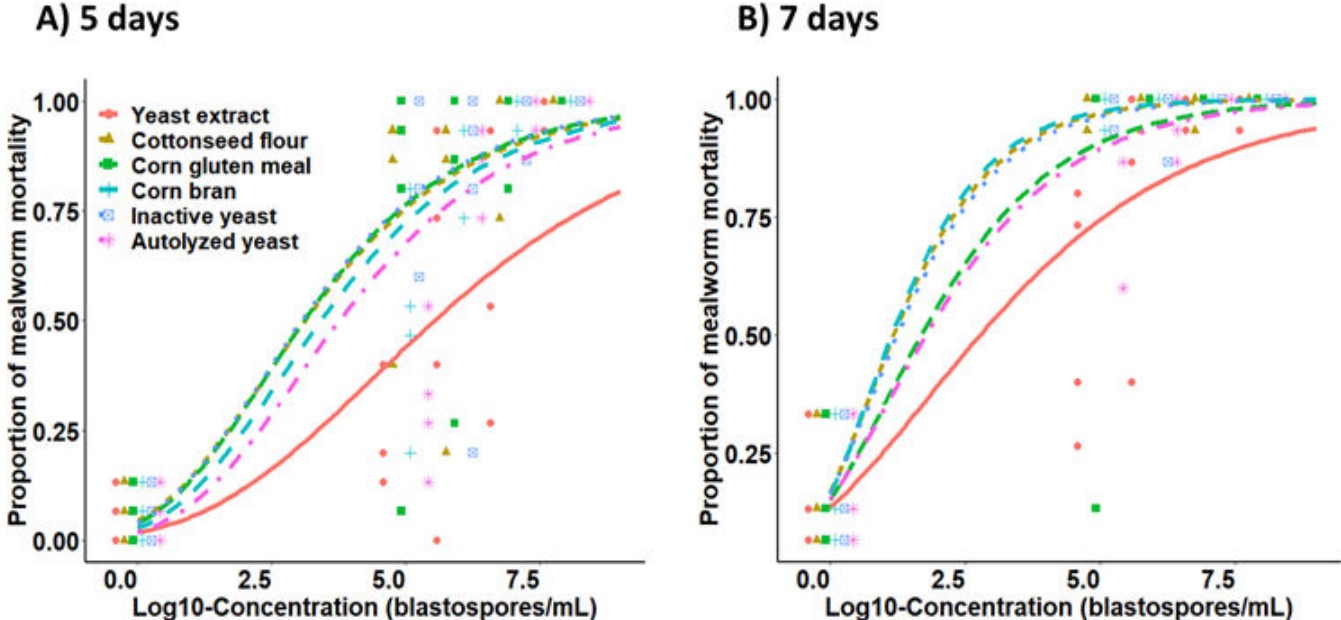

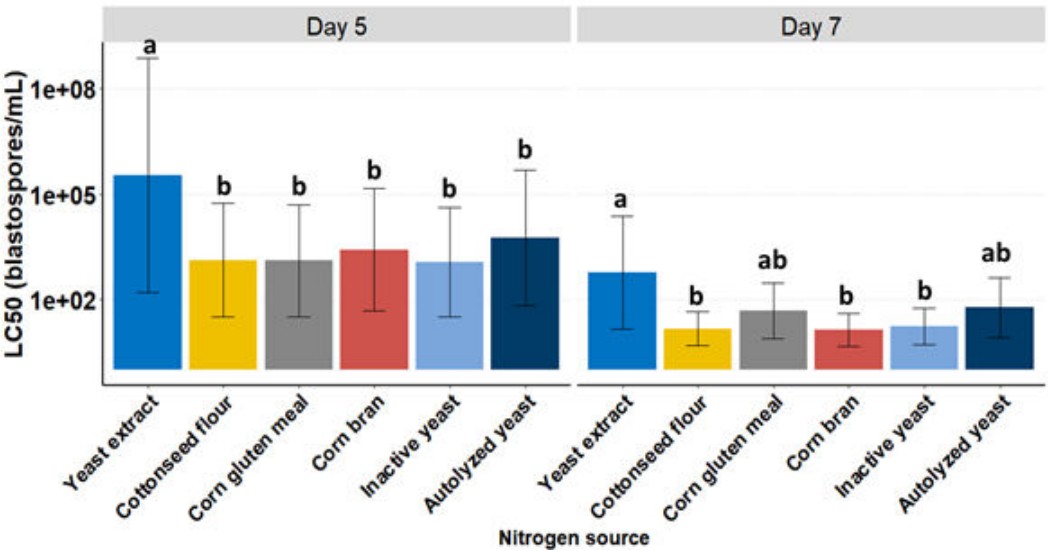

**FIG 5** Effect of complex nitrogen sources used to culture *Beauveria bassiana* s.l. IP 361 on the estimated application rates of blastospores required to kill 50% (LC$_{50}$) of *Tenebrio molitor* larvae. (A and B) Mortality-concentration data fitted to two-parameter Gompertz model at 5 and 7 days of application (symbols represent original data, whereas lines are fitted curves). (C) Estimated median lethal concentrations (LC$_{50}$, viable blastospores/mL) across nitrogen sources after 5 and 7 days of application. Error bars are 95% confidence limits. A representative experiment of four independent replications, where LC$_{50}$ values followed by the same letters indicate non-statistical differences ($P < 0.05$).

model, irrespective of nitrogen sources. At day 5 post-infection, the blastospores derived from cultures supplemented with cottonseed flour, corn bran, corn gluten meal, inactive yeast, or autolyzed yeast killed more mealworm larvae (~59- to 296-fold lower LC$_{50}$) than those blastospores grown with yeast extract (Fig. 5B). At day 7 post-infection, however, higher mortality rates (~35- to 43-fold lower LC$_{50}$) were reported for larvae treated with blastospores produced in culture media containing cottonseed flour, corn bran, or inactive yeast compared to those obtained from medium supplemented with yeast extract.

## Nitrogen sources shape the endogenous C:N ratio in blastospore composition

The complex organic nitrogen sources added to the fermentation media significantly altered the endogenous carbon and nitrogen contents in *B. bassiana* blastospores (Fig. 6A). Despite the small range among samples (42.5–44.3 mg/100 mg dry matter), the carbon content in blastospores was significantly changed by the nitrogen sources used for fungal fermentation ($F = 6.307$, df = 5, 12, and $P = 0.0043$). Fermentation medium with either autolyzed yeast or inactive yeast afforded higher endogenous carbon content in blastospores than cottonseed flour. Likewise, nitrogen sources significantly impacted the nitrogen content in blastospores ($F = 76.31$, df = 5, 12, and $P < 0.0001$), despite the small range of variation recorded among samples (4.46–7.05 mg/100 mg dry matter). Corn gluten meal and yeast extract contributed to higher nitrogen content in blastospores than autolyzed yeast, corn bran, and cottonseed flour. Furthermore, the C:N ratio in blastospores' composition was significantly influenced by all six complex organic nitrogen sources tested ($F = 101.1$, df = 5, 12, and $P < 0.0001$) (Fig. 6B). Autolyzed yeast and corn bran were the two sources that produced blastospores with the highest C:N ratio (i.e., 9.6:1 and 9.33:1, respectively), demonstrating that these cells contained more carbon than nitrogen content in dry matter. In contrast, the lowest endogenous C:N ratio was found in blastospores grown with corn gluten meal or yeast extract (6.1:1 and 6.57:1, respectively) due to their highest nitrogen content, which was expected as these sources also presented the lowest C:N ratios.

## Correlations between nitrogen sources and blastospore phenotypic traits revealed via multivariate analysis

A multivariate analysis, as depicted in a PCA biplot, effectively captured approximately 83.6% of the total variance within two principal components. This analysis unveiled significant correlations between the phenotypic traits of *B. bassiana* blastospores and the nitrogen sources investigated for their production (see Fig. 7). Blastospores cultivated in a medium supplemented with cottonseed flour displayed a notably high intracellular C:N ratio, elevated production yields, robust virulence traits, and heightened thermotolerance. However, this analysis indicated that cottonseed flour rendered blastopores less tolerant to UV-B radiation, although it was previously shown there were no significant differences among nitrogen sources (see Fig. 3B). Similar trends were observed in blastospores produced with inactive yeast and autolyzed yeast, albeit at varying magnitudes. In contrast, blastospores derived from liquid cultures supplemented either with yeast extract or corn gluten meal demonstrated the weakest virulence, reduced production yield, and lowest endogenous cellular C:N ratio. In summary, blastospores with higher endogenous C:N ratio (>6) demonstrated positive correlations with increased production yields, heat tolerance, and enhanced virulence, as evidenced by lower $LT_{50}$ and $LC_{50}$ values. Conversely, blastospores with a lower endogenous C:N ratio, while tending to exhibit less susceptibility to UV-B radiation, displayed elevated $LT_{50}$ and $LC_{50}$ values, indicating weaker virulence.

## Scale-up production of blastospores with two nitrogen sources

The growth kinetics of *B. bassiana* blastospores in submerged liquid cultures supplemented with autolyzed yeast or cottonseed flour in a 1-L benchtop bioreactor are shown in Fig. 8. As reported above, these two complex nitrogen compounds were previously selected in a screening test with various proteinaceous substrates. The growth pattern of *B. bassiana* with both autolyzed yeast and cottonseed flour followed a second-order polynomial regression with a significant *P*-value. Comparatively, cultures grown in medium with cottonseed flour displayed a faster growth rate, as measured by the slope, than cultures grown with autolyzed yeast. Blastospores' concentrations across time intervals were similar for cultures grown either with cottonseed flour or autolyzed yeast. On the last day of fermentation (i.e., 72 h post-inoculation), the highest mean

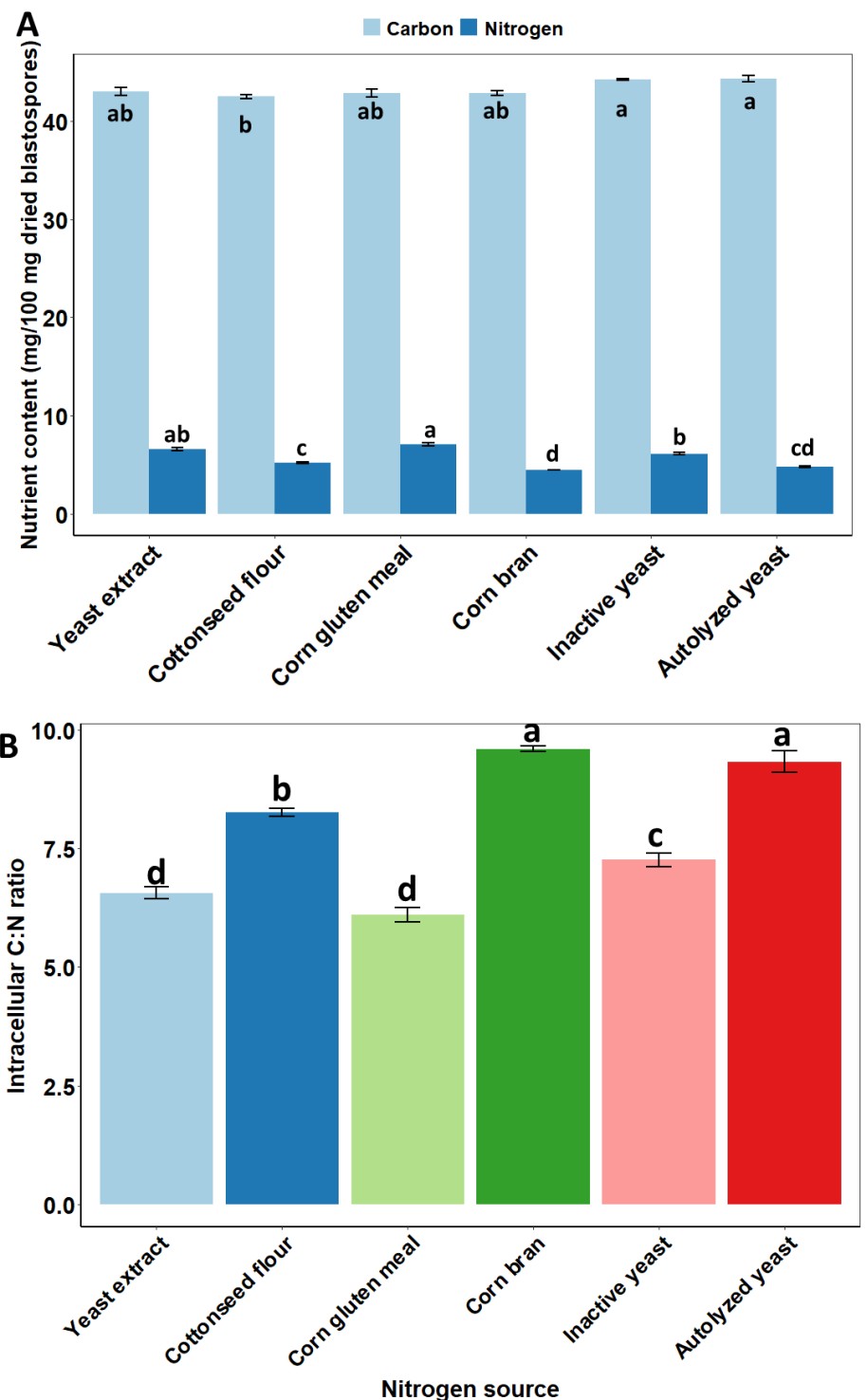

**FIG 6** Compositional cellular analysis based on nitrogen and carbon contents in blastospores' dry matter of *Beauveria bassiana* s.l. IP 361 produced in liquid culture media with different organic complex nitrogen sources. (A) Nutrient content (carbon and nitrogen) in blastospores' dry matter. Bars (mean values with standard errors, ±SE) with the same letter within the same nutrient content do not differ significantly at $P < 0.05$. (B) Endogenous carbon-to-nitrogen ratio in blastospores' dry matter. Bars (mean ± SE) with the same letter do not differ significantly at $P < 0.05$.

blastospore production in the liquid medium with cottonseed flour or autolyzed yeast reached $2.15 \times 10^9$ and $1.71 \times 10^9$ blastospores/mL, respectively. In general, both production curves reveal comparable growth kinetics for blastospore production with

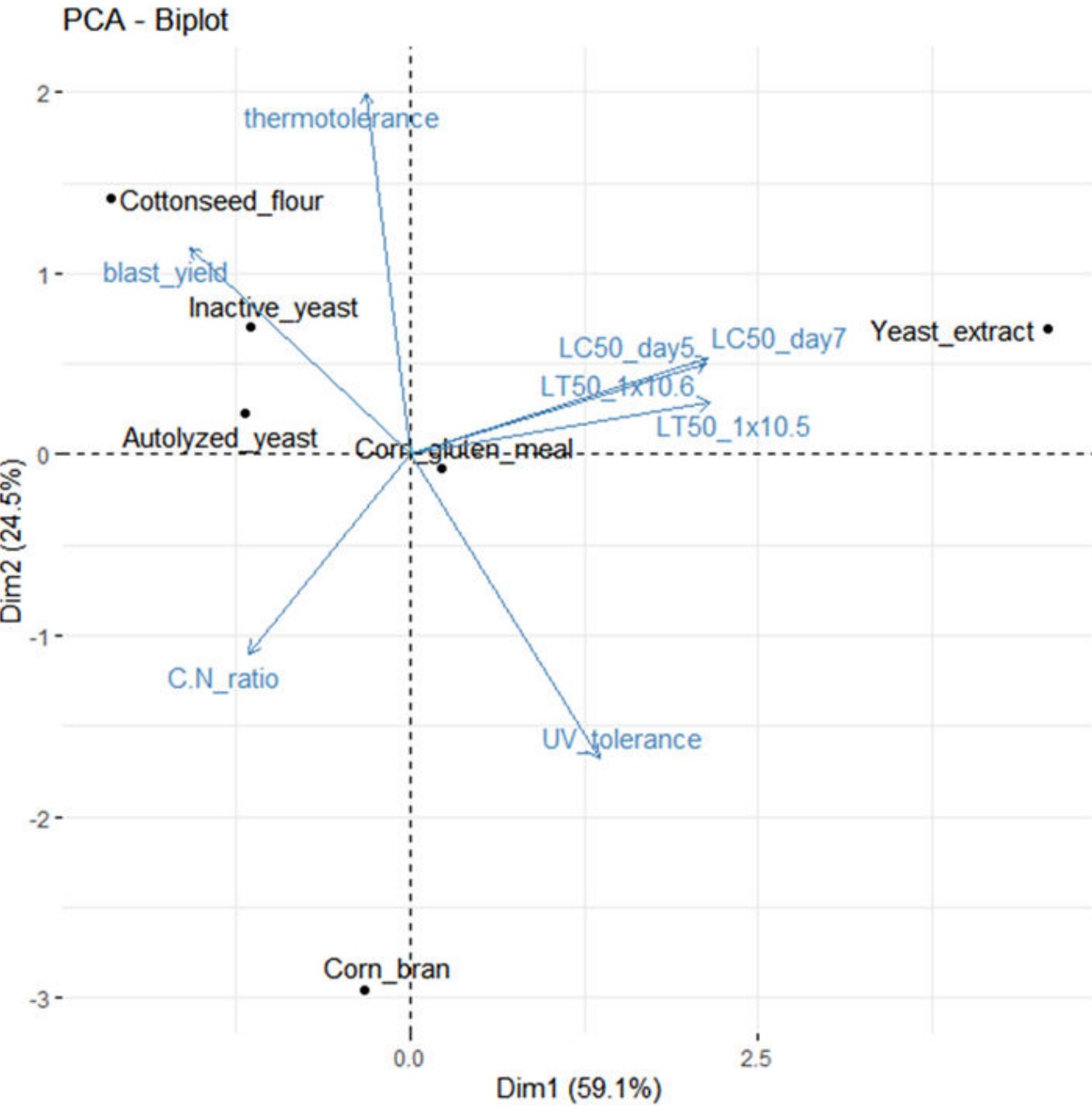

**FIG 7** Principal component analysis of the first two principal components, depicting the relationship among phenotypic traits of *Beauveria bassiana* blastospores and the nitrogen sources used to produce them. The first (PC1) and second (PC2) principal components explain 83.6% of the variance. Points that are close together correspond to observations that have similar scores on the components displayed in the plot. Vectors that point in the same direction correspond to variables with high correlation, and the length of the vector indicates its magnitude.

both nitrogen sources tested ($F = 0.96$, df = 2, 44, and $P = 0.39$). This suggests that these nitrogen substrates are equally effective for blastospore production by this *B. bassiana* strain (Fig. 8A).

By indirectly measuring glucose levels in the spent medium, the glucose uptake was more pronounced with fungal cultures grown with cottonseed flour rather than autolyzed yeast in order to sustain higher blastospore yields ($F = 5.65$, df = 1, 44, and $P = 0.02$). However, fungal cultures demonstrated a similar glucose consumption rate irrespective of the nitrogen source tested ($F = 1.63$, df = 1, 44, and $P = 0.21$). Throughout

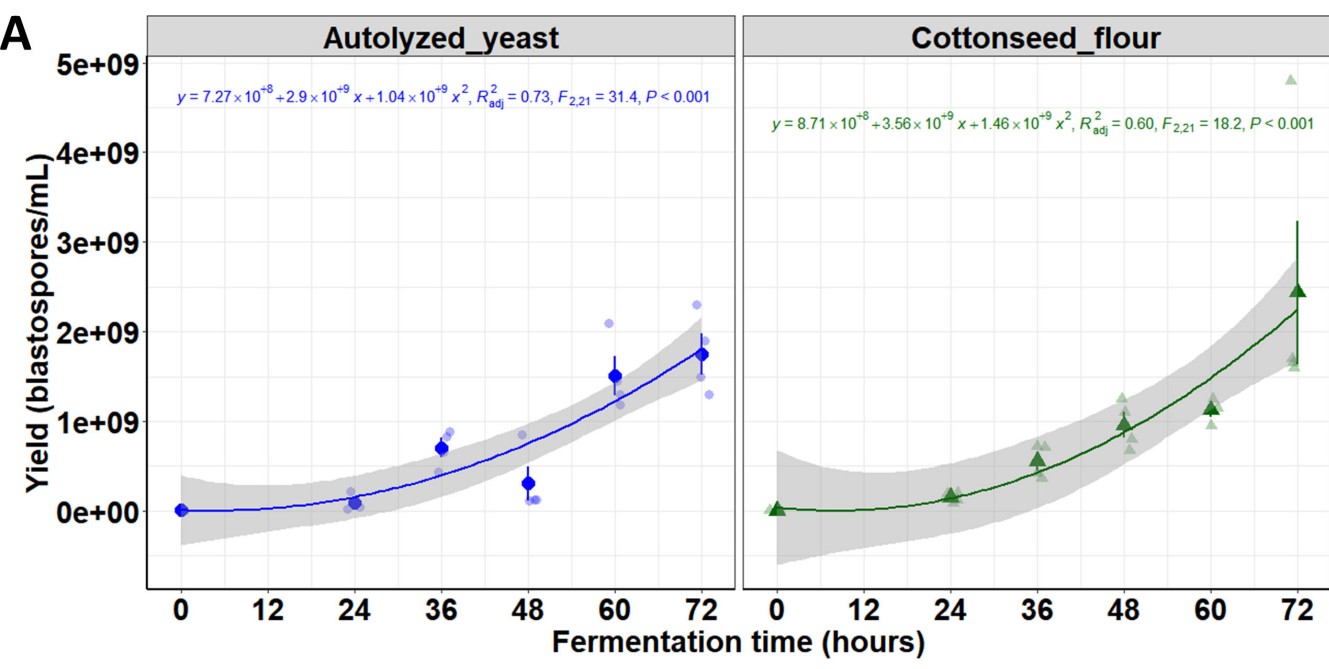

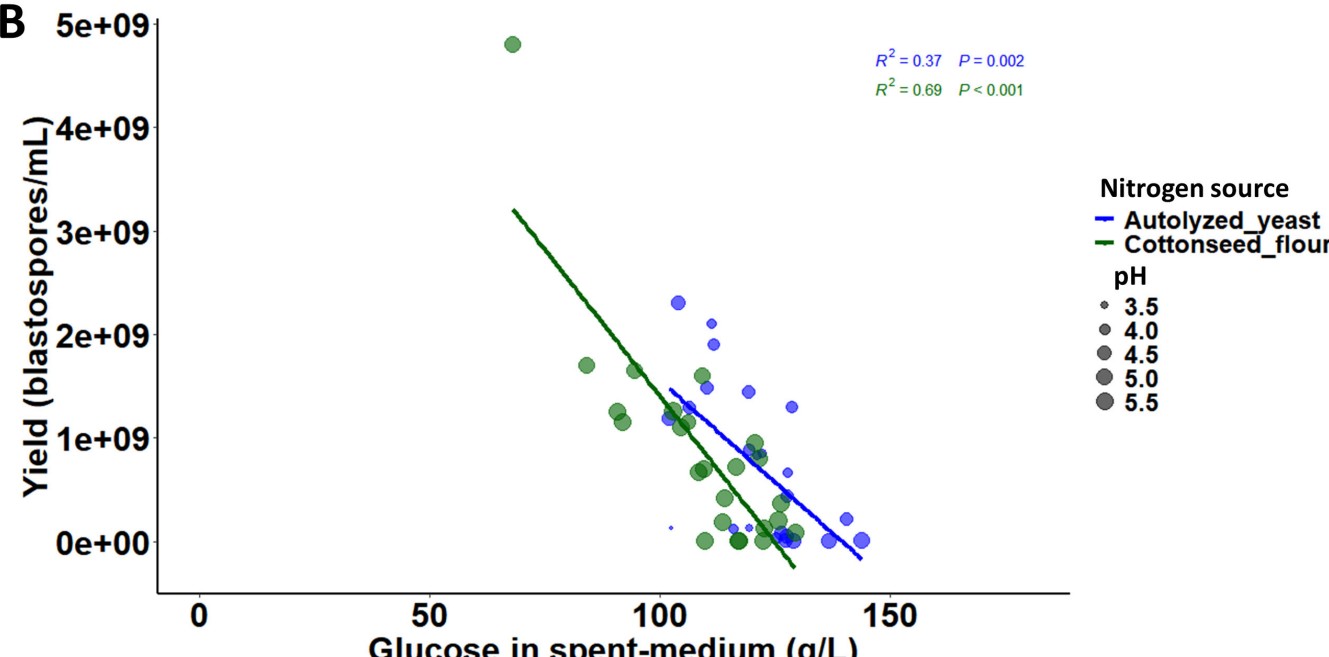

**FIG 8** Impact of two low-cost complex nitrogen sources (cottonseed flour and autolyzed yeast) on the growth kinetics of submerged liquid cultures of *Beauveria bassiana* s.l. IP 361 conducted in a 1-L benchtop bioreactor. (**A**) Solid lines represent the second-order polynomial regression curves accompanied by shaded 95% confidence intervals. Symbols (circle and triangle) represent the mean values with standard errors (±SE) along with observational data (shaded points or triangles). Second-order polynomial equations followed by adjusted $R^2$, $F$-statistic, and $P$-value are displayed within each plot. (**B**) Relationship between blastospore yield and remaining glucose in the spent medium represented by fitted linear solid lines (adjusted $R^2$ and $P$-value) accompanied by pH of the culture broth (the bigger the circles, the higher the pH).

the fermentation course, the pH of the submerged cultures displayed greater fluctuations in the presence of autolyzed yeast (Fig. 8B). It decreased from 4.8 at the start to 3.56 at 48 h, stabilizing at 4.35 after 72 h. In contrast, cultures grown with cottonseed flour exhibited a subtle pH variation during submerged growth, initiating at 5.43 and dropping to 5.15 by the completion of the fermentation process.

## Shelf life and bioactivity of blastospores during cold storage

Regardless of the nitrogen source tested in the fermentation medium and the length of time during storage, blastospores killed 100% of mealworm larvae between 5 and 7 days post-treatment, and survival rates in fungal treatments significantly decreased in relation to the control groups in all cases ($\chi^2$ = 2.56, df = 9, and $P$ < 0.0001). When mealworms were treated with stored *B. bassiana* blastospores produced in culture media supplemented with different nitrogen sources, the survival rates were similar for periods of 0, 1, 4, and 24 weeks of storage ($P$ > 0.05). On the other hand, blastospores produced with cottonseed flour resulted in lower survival rates, accompanied by lower $LT_{50}$ values, than the blastospores derived from autolyzed yeast for periods of 2, 8, and 12 weeks ($P$ < 0.05) (Fig. 9). In most cases, in the control groups, more than 95% of mealworm larvae survived.

The assessment of the shelf life of air-dried blastospores unveiled a substantial impact of nitrogen sources on their storage stability. Notably, blastospores generated with cottonseed flour exhibited prolonged viability during storage, surpassing the retention of viability compared to those produced with autolyzed yeast after 180 days (6 months) at 4°C. Blastospores grown with cottonseed flour displayed consistently high viability levels, maintaining over 98% cell viability throughout the entire 180-day storage period. In contrast, blastospores produced with autolyzed yeast experienced a decline in viability from an initial 100% to 89.5% during storage (Fig. 10).

## DISCUSSION

After carbon and oxygen, nitrogen is an essential macronutrient that is required for the growth of fungi and plays an important role in the regulation of secondary metabolism. Usually, fungi can utilize a wide variety of nitrogen sources present in the environment (44). Here, we demonstrated how plant-based or microbial nitrogen substrates derived from agro-industrial byproducts added to a liquid fermentation medium impacted the production of *B. bassiana* blastospores, their tolerance to heat and simulated UV-B radiation, their virulence toward a specific host insect, and shelf life under cold storage followed by the assessment of their insecticidal bioactivity during storage. Despite the slight differences in their C:N ratios, these proteinaceous compounds also played a crucial role in altering the endogenous C:N ratio in blastospores, which could explain the notable differences in their virulence, quality, fitness, and shelf life during storage.

The use of agro-industrial byproducts, which are rich in protein content but also in other nutrients (e.g., carbohydrates, fatty acids, vitamins, and minerals), can significantly reduce the total cost of the liquid medium while simultaneously improving the massive blastospore production and the ecological fitness of this fungal propagule. For example, the commercially available yeast extract is the conventional nitrogen source widely employed for cultivating filamentous fungi. It is known for its elevated nitrogen content and is considered the gold standard substrate in fungal fermentation. However, this source tends to be associated with higher costs. As an alternative, we conducted a comparison of various organic complex nitrogen compounds against the standard commercial yeast extract (see Table 1). We explored options for reducing production costs while ensuring that the quality and ecological fitness of blastospores remained unaffected. Understanding fungal nutrition and physiology can guide future studies in establishing an appropriate fermentation medium for producing high-quality and robust blastospores for use in pest biocontrol programs.

Noteworthy is the nutritional versatility of *B. bassiana* in utilizing a broad range of nitrogen sources, which facilitates its colonization and occupation of diverse host niches, supporting its pathogenicity, saprophytic, and endophytic lifestyles (45). In this sense, *B. bassiana* is armored with an enzymatic machinery capable of metabolizing and assimilating complex organic nitrogen sources. Interestingly, *B. bassiana* and *Metarhizium robertsii*, another generalist entomopathogenic and endophytic fungus, perform much better when growing in liquid media containing complex proteinaceous substrates

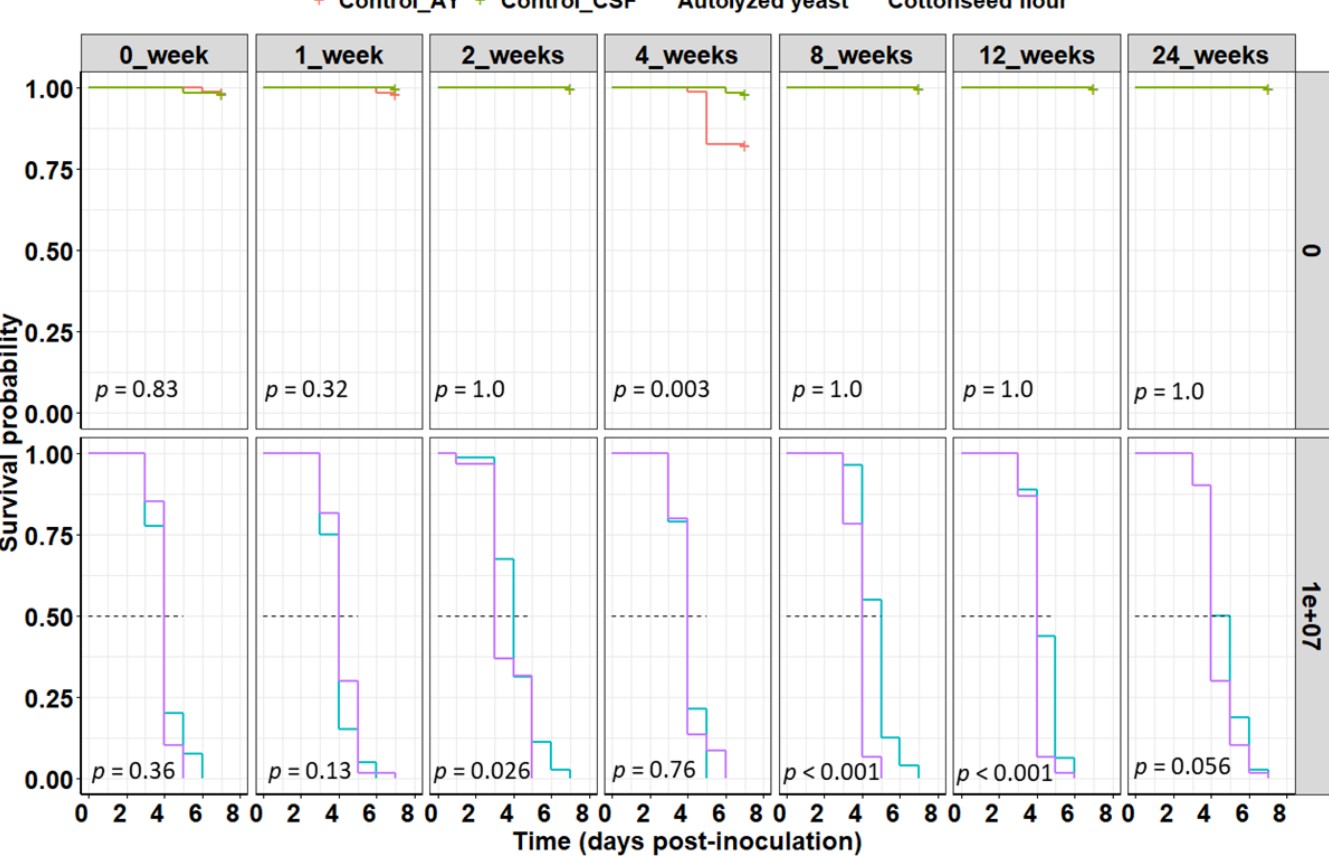

**FIG 9** Impact of the nitrogen sources (cottonseed flour and autolyzed yeast) on the bioactivity of air-dried blastospores of *Beauveria bassiana* s.l. IP 361 for up to 24 weeks of cold (4°C) storage. Survival plots for *Tenebrio molitor* larvae exposed to blastospores ($1 \times 10^7$ viable blastospores/mL) derived from fermentation with different nitrogen sources or not exposed to fungal treatments (*i.e.*, control groups indicated by "0"). Significant differences in the survival curves between the two treatment groups (nitrogen sources) or between the two control groups are shown for each cold storage period at $P < 0.05$. Legends: Control_AY = control for autolyzed yeast, Control_CSF = control for cottonseed flour.

rather than simple (mineral or organic) sources leading to maximum blastospore yields within 2–3 days (11, 24). Despite all these phenotypic characterization studies, the carbon and nitrogen metabolism pathways remain poorly understood in *B. bassiana* blastospores grown with different complex organic nitrogen sources, which warrants additional research.

The potential utilization of agro-industrial wastes or byproducts as sustainable and cost-effective feedstocks in microbial fermentation media has gained considerable attention in recent years (46). These proteinaceous byproducts are not only economical but also rich in nutrients and various chemical compounds, making them a viable and cost-effective resource for producing microbial agents (47, 48). Additionally, incorporating these substrates into the production process contributes to more sustainable practices by reducing environmentally persistent elements with slow degradability (49). This approach adds value by transforming low-cost agro-industrial byproducts into valuable environmentally-friendly biopesticides using the context of biorefineries. Leveraging filamentous fungi for the bioconversion of organic byproducts into green biopesticides aligns with the principles of sustainability within the context of the circular economy. This not only mitigates the need to use cereal grains for human food and livestock feed but also represents a noble and feasible approach to achieving sustainability in agricultural practices.

One of the primary objectives when utilizing agro-industrial byproducts as feedstocks for *B. bassiana* blastospore production is to achieve a substantial yield of propagules in a

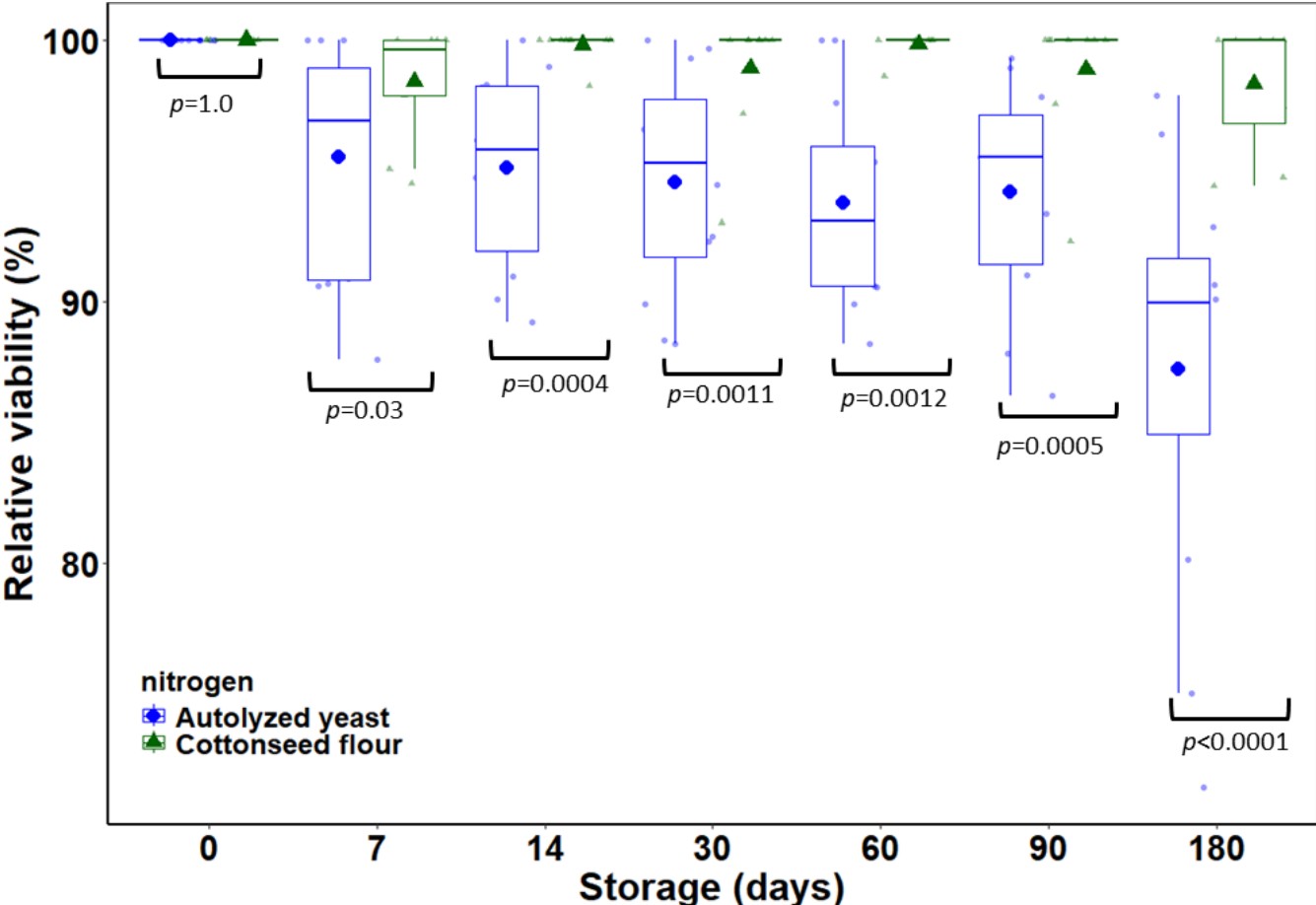

**FIG 10** Impact of the nitrogen sources (cottonseed flour and autolyzed yeast) on the shelf life of air-dried blastospores of *B. bassiana* s.l. IP 361 for up to 180 days (6 months) of cold (4°C) storage. The relative viability of stored blastospores was calculated in relation to the germination of blastospores assessed at time zero (right after drying). The central line of each box plot represents the median relative viability, tthe box boundaries indicate the interquartile range (IQR), and the whiskers extend to the furthest points within 1.5 times the IQR from the box. Large filled circles and triangles represent mean values, while small shaded points and triangles represent observational data. Significant differences in the relative viability of blastospores derived from fermentation with cottonseed flour or autolyzed yeast are indicated by *P*-value < 0.05.

cost-effective manner without compromising quality. Our findings indicate that all tested nitrogen sources yielded blastospores at levels comparable or even superior to yeast extract. In this study, *B. bassiana* cultured in a liquid fermentation medium with cotton-seed flour or autolyzed yeast using a benchtop bioreactor resulted in high blastospore yields (>1 × 10^9/mL) within 72 h of cultivation. Interestingly, based on our results from shake flask media experiments, inactive yeast or corn gluten meal also exhibited the potential to provide similarly high blastospore yields. As reported by Mascarin et al. (11), cottonseed flour and autolyzed yeast have been identified as effective components for optimizing yeast-like production in certain *B. bassiana* strains. However, given the significant genetic diversity among entomopathogenic fungi, further investigation is imperative for accurately assessing the impact of these nutritional components in screening new strain candidates aimed at biocontrol of arthropod pests. The exploration of liquid fermentation for *B. bassiana* and other fungal biocontrol agents may present a viable, eco-friendly, and cost-effective alternative to the more commonly practiced solid-state fermentation (SSF) using cereal grains like rice. Adopting liquid fermentation for entomopathogenic fungi may also alleviate the waste generated from SSF after spore harvesting.

The survival and persistence of entomopathogenic fungi in agricultural crops hinge significantly on their susceptibility to elevated temperatures and ultraviolet (UV-B)

radiation from sunlight. This underscores the importance of systematic research into manipulative strategies involving nutrition and physiology to enhance the fitness and performance of fungal biocontrol agents. Notably, entomopathogenic fungi and their propagules exhibit considerable inter- and intra-specific variations in susceptibility to heat and UV-B radiation (50–53). The nutritional and environmental conditions during fungal growth, whether in solid or liquid cultures, play a pivotal role in influencing fungal tolerance to heat and UV-B radiation (54–59). This study highlights explicitly that the heat tolerance (45°C) of *B. bassiana* IP 361 was significantly influenced by the nitrogen source employed during liquid fermentation. Blastospores produced with cottonseed flour, autolyzed yeast, or inactive yeast exhibited similar survival rates after heat exposure to those produced with yeast extract. Conversely, no discernible effect of nitrogen sources on blastospore tolerance to UV-B radiation was observed, with all variants displaying similar UV-B tolerance. While not directly assessed in our current study, earlier research has indicated that the choice of nitrogen source in the culture medium significantly affects desiccation tolerance and the shelf life of cold-stored blastospores of *B. bassiana* and *Cordyceps fumosorosea* (formerly *Isaria fumosorosea*) (11, 16). Assessing the impact of nutrients, such as nitrogen, on abiotic stress tolerance of blastospores, especially heat and UV-B, is key in the context of global warming, as the success of fungal biopesticides in controlling arthropod pests is also dependent on conducive environmental factors (19).

The endogenous nitrogen content in blastospores appears to be notably influenced by the nitrogen source supplemented in the culture medium, and this consequently affects the intracellular C:N ratio. Cliquet and Jackson (23) highlighted a similar observation, noting that elevated levels of endogenous nitrogen content in blastospores of *C. fumosorosea*, produced in a nitrogen-rich culture medium, conferred substantial tolerance to desiccation through air drying. Additionally, *C. fumosorosea* blastospores exhibited the highest mannitol-to-trehalose ratio when cultivated in a medium enriched with 5 g/L soya peptone, suggesting the importance of this ratio for improved heat tolerance in this propagule (25). In our study, the incorporation of complex organic nitrogen sources into the fermentation media led to significant alterations in the carbon and nitrogen contents of *B. bassiana* blastospores. Notably, autolyzed yeast and corn bran produced blastospores with the highest intracellular C:N ratios, whereas the lowest ratios were achieved with corn gluten meal and yeast extract, indicating higher intracellular nitrogen content. This finding underscores the impact of nitrogen sources on the endogenous nutritional composition of blastospores, shedding light on potential avenues for optimizing their physiological traits.

Another crucial parameter to consider during the downstream processing stage in the development of blastospore-based biopesticides is the shelf life. Our data align with previous studies that have demonstrated the impact of nitrogen sources, such as casamino acids, cottonseed flour, and corn steep liquor combined with or without yeast extract on the shelf stability of non-refrigerated and refrigerated blastospores of *B. bassiana* and *M. robertsii* (29, 30, 60). Therefore, while the genetic component of the fungal strain undoubtedly plays a role in shelf life, the composition of the nutritional media is also critical for maintaining the viability of blastospores under stored conditions, as nitrogen and other nutrients present in these proteinaceous substrates may affect this phenotypic trait.

The topical bioassay conducted with mealworm larvae, representing the natural route of infection for *B. bassiana* through penetration of the insect cuticle, aimed to assess the time course of infection against larvae treated with fungal blastospores produced in fermentation media featuring different organic nitrogen sources. Our findings consistently identified inactive yeast, corn bran, and cottonseed flour as particularly suitable for producing highly virulent blastospores compared to other nitrogen sources tested. The virulence of *B. bassiana* blastospores toward mealworm larvae was indeed influenced by the nitrogen source, with blastospores produced in media containing inactive yeast, corn bran, or cottonseed flour displaying higher virulence, as evidenced by lower

$LC_{50}$ and $LT_{50}$ values than blastospores produced with other nitrogen sources. While prior studies have documented the influence of culture medium composition on the virulence of entomopathogenic fungi against arthropod pests (27, 28, 61), our study uniquely reveals the significant impact of nitrogen sources on the virulence of *B. bassiana* blastospores against an insect host. In many filamentous fungi, regulating carbon and nitrogen metabolic pathways involves carbon catabolite repression and nitrogen catabolite repression. These regulatory mechanisms are often tightly linked to crucial aspects such as virulence, morphogenesis (i.e., dimorphic growth), and stress responses in pathogenic fungi (62). However, the relationship between nutrient metabolism regulation and virulence of *B. bassiana* blastospores cultivated in diverse, complex organic nitrogen substrates is poorly characterized and a subject of interest. For instance, the BbAreA GATA-like transcription factor in *B. bassiana* has been identified as a key activator of genes involved in the utilization of a wide range of nitrogen sources. Additionally, it plays a role in modulating insect virulence and plant root colonization (45). In this sense, the relationship of different nitrogen sources employed to grow *B. bassiana* with the carbon/nitrogen catabolite repression system and its contribution to dimorphic growth, insect virulence, stress responses, and shelf life of blastospores remain to be fully deciphered.

Nitrogen serves as a fundamental nutrient for fungal growth, playing a crucial role in various metabolic pathways such as protein synthesis, cellular components' elaboration, and secondary metabolite production. The levels, compositions, and gene expressions of antioxidant and cuticle-degrading enzymes, as well as several secondary metabolites produced by *B. bassiana* are indeed influenced by nitrogen metabolism, which, in turn, is highly responsive to the nitrogen sources provided for the fungal growth (44, 63, 64). Further investigations are warranted to delineate the specific metabolic pathways that influence the virulence of entomopathogenic fungi based on the nitrogen source used to supplement the culture medium. The identified genes may serve as valuable molecular markers of virulence in entomopathogenic fungi mediated by nitrogen sources. Proteomics can also prove beneficial in elucidating the protein profiles in blastospores produced with different nitrogen sources, providing insights into specific proteins linked to increased production yields, virulence, shelf stability, and tolerance to abiotic stresses. These molecular and proteomic approaches hold the potential to enhance our understanding of the intricate relationship between nitrogen metabolism and the multifaceted traits critical for the effectiveness of entomopathogenic fungi in biological pest control.

Nitrogen plays a crucial role in the composition of fermentation media, influencing fungal metabolism and, consequently, the quality and fitness of produced blastospores. Particularly noteworthy are the vitamins and minerals inherent in complex organic proteins, potentially eliminating the need for external supplementation. Our results emphasize the significance of carefully selecting nitrogen sources from agro-industrial byproducts for cost-effective, high-quality blastospore production in entomopathogenic fungi, using *B. bassiana* as a model organism. These findings align with previous studies (11, 65), highlighting the importance of diverse nitrogen sources in blastospore production, fitness, and quality. Contrary to the perceived sensitivity of thin-walled blastospores to abiotic stresses compared to aerial conidia, this study, supported by our previous results (50), underscores the pivotal influence of different genotypes (strains) and nutrients during liquid fermentation in modulating blastospore tolerance to UV radiation and heat shock stresses. These insights pave the way for designing strategies that integrate genetics and nutrition to cultivate more ecologically fit blastospores for effective pest control.

Overall, these findings are pivotal for crafting media formulations capable of yielding high-quality and abundant blastospores with enhanced virulence and multi-stress tolerance in entomopathogenic fungi. Such advancements are integral to the development of innovative and eco-friendly mycopesticides. The envisioned production system should prioritize cost-effectiveness and efficiency, leveraging nutrient sources

such as value-added agro-industrial byproducts in media formulation. This approach aligns seamlessly with the principles of a green circular economy and contributes to a sustainable environment and economy with less use of chemical inputs in agricultural ecosystems. Importantly, given that nutritional components can represent a high cost in microbial product production (66), our study underscores the significance of exploiting low-cost agro-industrial byproducts for mass-producing mycopesticides based on blastospores obtained through submerged liquid fermentation technology. This approach is poised for further integration into sustainable biological pest control programs.

## Conclusion

This study offers unprecedented insights into the effects modulated by complex nitrogen sources on the production performance, endogenous C:N ratio, abiotic multi-stress tolerance, virulence, and shelf stability after dehydration of *B. bassiana* blastospores obtained by submerged liquid fermentation. Furthermore, our study showcases, through a lab-scale benchtop bioreactor, that readily available and cost-effective vegetable-based protein sources obtained from agro-industrial processes are well-suited for the industrial-scale production of these propagules and can replace yeast extract, a high-cost nitrogen source. This approach yields exceptional quantities within short fermentation times, typically ranging from 2 to 3 days. In light of these observations and consistent with prior studies (11, 20, 21, 29, 30), we argue that cottonseed flour stands out as one of the most suitable protein sources for enhancing *B. bassiana* blastospore mass production by submerged fermentation with an impact on their endogenous C:N ratio that correlates with enhanced ecological fitness, insect virulence, and extended shelf life. Notwithstanding the inherent genetic diversity among *B. bassiana* strains, our data further endorse the utility of cottonseed flour for blastospore production across multiple strains of this fungus. Finally, our integrative approach, combining insights into fungal physiology and nutrition with the manipulation of desired phenotypic traits, is applicable to various fungal biopesticides, including *Akanthomyces* spp., *Metarhizium* spp., *Hirsutella* spp., *Simplicillium* spp., *Purpureocillium lilacinum*, *Aschersonia* spp., and *Cordyceps* spp. This fermentation strategy has the potential to broaden the repertoire of high-quality blastospore-based products globally available.

## Highlights

- Influence of nitrogen sources on biological and ecological traits in *B. bassiana* blastospores.
- Nitrogen sources dictate the virulence and multi-stress tolerance of blastospores.
- Nitrogen sources shape endogenous cellular C:N ratio in blastospores.
- Cottonseed flour improves ecological fitness, shelf life, and virulence of blastospores.

### ACKNOWLEDGMENTS

This research was supported by grants from the Fundação de Amparo à Pesquisa do Estado de Goiás (FAPEG, 201710267000515) and Instituto Nacional de Ciência e Tecnologia em Entomologia Molecular, Universidade Federal do Rio de Janeiro, Rio de Janeiro, Brazil (INCT-EM, 465678/2014-9). The Brazilian National Council for Scientific and Technological Development (CNPq) provided a Ph.D. scholarship for V.H.L., an undergraduate research grant for A.T.M., and the grants PQ 755 306319/2018-7 and 308375/2022-0 for É.K.K.F. The publication fee was covered by a grant from the Escola de Veterinária e Zootecnia at Universidade Federal de Goiás (EVZ/UFG), Fundação de Apoio à Pesquisa (EVZ/FUNAPE; CC 11254).

## AUTHOR AFFILIATIONS

[1]Programa de Pós-graduação em Ciência Animal, Universidade Federal de Goiás, Goiânia, Goiás, Brazil

[2]Instituto de Patologia Tropical e Saúde Pública, Universidade Federal de Goiás, Goiânia, Goiás, Brazil

[3]Laboratório de Microbiologia Ambiental, Embrapa Meio Ambiente, Jaguariúna, São Paulo, Brazil

## AUTHOR ORCIDs

Valesca Henrique Lima  http://orcid.org/0000-0002-3818-0508
Alexandre Toshihiro Matugawa  http://orcid.org/0009-0005-5712-6004
Gabriel Moura Mascarin  http://orcid.org/0000-0003-3517-6826
Éverton Kort Kamp Fernandes  http://orcid.org/0000-0001-7062-3295

## FUNDING

| Funder | Grant(s) | Author(s) |
| --- | --- | --- |
| Fundação de Amparo à Pesquisa do Estado de Goiás (FAPEG) | 201710267000515 | Éverton Kort Kamp Fernandes |
| Instituto Nacional de Ciencia e Tecnologia em Entomologia Molecular, UFRJ | 465678/2014-9 | Éverton Kort Kamp Fernandes |
| Conselho Nacional de Desenvolvimento Científico e Tecnológico (CNPq) | PQ 306319/2018-7 | Éverton Kort Kamp Fernandes |
| Conselho Nacional de Desenvolvimento Científico e Tecnológico (CNPq) | PQ 308375/2022-0 | Éverton Kort Kamp Fernandes |
| Escola de Veterinária e Zootecnia, Universidade Federal de Goiás / Fundação de Apoio à Pesquisa | EVZ/FUNAPE; CC 11254 | Éverton Kort Kamp Fernandes |

## ETHICS APPROVAL

Access to Brazilian genetic heritage was approved by the Genetic Heritage Management Council (CGen) of Brazil (SisGen protocol #A420934).

## ADDITIONAL FILES

The following material is available online.

Open Peer Review

**PEER REVIEW HISTORY (review-history.pdf).** An accounting of the reviewer comments and feedback.

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
