## [Reviewer comments · Microbiology Spectrum]

Microbiology Spectrum

Complex Nitrogen Sources from Agro-Industrial Byproducts: Impact on Production, Multi-Stress Tolerance, Virulence, and Quality of *Beauveria bassiana* blastospores

Valesca Lima, Alexandre Matugawa, Gabriel Mascarin, and Everton Fernandes

Corresponding Author(s): Gabriel Mascarin, Empresa Brasileira de Pesquisa Agropecuaria

Review Timeline:

Submission Date:	November 26, 2023
Editorial Decision:	January 23, 2024
Revision Received:	February 2, 2024
Editorial Decision:	March 18, 2024
Revision Received:	March 21, 2024
Accepted:	March 28, 2024

Editor: Lea Atanasova

Reviewer(s): Disclosure of reviewer identity is with reference to reviewer comments included in decision letter(s). The following individuals involved in review of your submission have agreed to reveal their identity: Travis Glare (Reviewer #2)

Transaction Report:

DOI: <https://doi.org/10.1128/spectrum.04040-23>

Re: Spectrum04040-23 (Optimizing *Beauveria bassiana* Blastospores: Impact of Complex Nitrogen Sources from Agro-Industrial Byproducts on Production, Multi-Stress Tolerance, and Virulence)

Dear Dr. Gabriel Moura Mascarin:

Thank you for submitting your manuscript to Microbiology Spectrum. Based on the reviewers comments you are invited to submit a revised version of your manuscript. Please carefully address the issues raised by the reviewers as well as the following issue for me (editor):

"The manuscript is written as if nitrogen was the sole driver of the differences, ... the emphasis of the paper is wrong."

When submitting the revision, please provide (1) point-by-point responses to the issues raised by the reviewers as file type "Response to Reviewers," not in your cover letter, and (2) a PDF file that indicates the changes from the original submission (by highlighting or underlining the changes) as file type "Marked Up Manuscript - For Review Only".

Revision Guidelines

Sincerely,
Lea Atanasova
Editor
Microbiology Spectrum

Reviewer #2 (Comments for the Author):

This was a well written and interesting manuscript on the effect of different complex nitrogen sources on blastospore production of *B. bassiana*, viability under different conditions and virulence. The data presented and the research approach were excellent and I found it a very interesting and useful study. I have a few points though. The manuscript is written as if nitrogen was the sole driver of the differences, although in several parts it is acknowledged that the sources are complex and contain other compounds that will affect the blastospores (e.g. line 644 onwards). Given the C:N ratios of the sources, it really isn't just nitrogen. This does not detract from the results, but the emphasis of the paper is wrong.

The authors talk about the contribution of media to the cost of production, but provide no cost comparisons of their own different sources. The medium composition in the paper is quite expensive regardless of the nitrogen source in my opinion.

Is 40C an appropriate storage temperature when companies are looking for shelf life at room temperature?

There was no variation in the % in the media of the nitrogen source- has this been looked at previously?

Specific comments:

Line 103 *in vitro* in italics

Line 121-122- which reference covers which aspect? Especially which papers report blastospore infectivity or toxicity?

Line 179-180 Inoculated with spores to establish a concentration of blastospores? Do you mean final concentration of spores?

Line 208 what pore size of the cheesecloth?

Line 275 mycelia

Line 279 freeze-dried?

Line 310- Not clear what the volume of 2×10^{10} blastospores was? I assume this was straight from the fermenter. What was the loss through filtering at this step?

Line 321 does these air-dried blastospores mean with or without diatomaceous earth?

Line 375 what does "marked" mean here?

Line 497 what is the pronounced impact? Next sentence says similar?

Line 636 I wouldn't use the word "remarkable" as its not very specific nor accurate. Just higher than would be better

Line 644 It is not clear from the results that nitrogen was what was optimised, given the small differences in C:N ratio between some of the sources

Line 670 There is no comparisons mentioned in the text between conidia and blastospores, so hard to determine the comparative value of producing one over the other. Beyond the scope of this paper though.

Line 685 The 72% of the cost as media components is very unlikely, given the cost of labour in most countries and equipment depreciation etc. A table of comparative cost of each nitrogen source would have been useful if this is your argument.

Line 693 complex nitrogen source

Line 698 yields are good, not exceptional.

Line 768 space needed between types and and

Lines 796, 916 Italics for species

In this manuscript, authors provided experimental data to unveil exogenous nitrogen source affects fungal blastspores production, endogenous C:N ratio, virulence, shelf life, etc. Different cellular C:N ratio is an interesting view to explore the effect of nitrogen on fungal traits. However, the major concern is, this work seems to be a summary of phenotypical tests, without any solid evidence or clues investigate how the nitrogen supplements affects the microbial traits. On the other hand, cottonseed flour has already been proven to be a good nitrogen for *Beauveria bassiana's* growth in previous publication, so it's not an innovative finding. In my point of view, is not qualify enough to publish unless authors add results and discussion associate with the mechanism research. In addition, The C:N ration displayed in Table 1 doesn't match the data shown in figure. In Table 1, the C:N ratio of "cottonseed flour" is same as "corn gluten meal" with 5:1. It should be incorrect.

February 2, 2024

Microbiology Spectrum,
Editorial Office, ASM Journals

Manuscript #: Spectrum04040-23

Response to Reviews

Editor:

"The manuscript is written as if nitrogen was the sole driver of the differences, ... the emphasis of the paper is wrong."

Response: In this work, we used complex organic nitrogen compounds that, by their own nature, contain not only nitrogen but also other important nutrients to support fungal growth and metabolism, such as carbohydrates, vitamins, and minerals, due to their vegetable or microbiological origin. For example, yeast extract is rich in vitamin B complex. However, we cannot control the nutritional composition of such complex organic nitrogen sources, so it is impossible to have the same amount of nutrients in these nitrogen sources. In order to standardize and offer the minimum amount of minerals and vitamins in all culture media, we used our basal medium and glucose in all fermentation tests carried out in this study. Therefore, the only variable that we investigated was the nitrogen sources that came from different origins and manufacturers but were chosen due to their high total nitrogen content (> 6% total N of dry matter). Nitrogen is a critical and the most expensive nutrient in the fermentation media of filamentous fungi, so it is imperative to find inexpensive and abundant sources of this nutrient to make feasible the mass production of filamentous fungi for biocontrol of pests. Companies are still struggling to find the proper nitrogen source for fermentation optimization of their fungal strains, and it is of paramount importance to continue this kind of study to determine new and inexpensive sources of nitrogen compounds to offer them to the industry sector.

Even though we provided the C:N ratio of all nitrogen sources studied here, we found very small differences between nitrogen sources, which has no major impact on fungal phenotypic characteristics. Actually, these nitrogen sources impacted on the endogenous C:N ratio of blastospores produced, and we discussed this in our manuscript. Therefore, there is no need in discussing or emphasizing the C:N ratio of the nitrogen sources, but rather we did discuss and emphasize the impact that these nitrogen sources can have on the endogenous C:N ratio of blastospores and how this correlates with other phenotypic traits.

In addition, as we used glucose at a high rate (14%), *B. bassiana* – like many other filamentous fungal species – has a strong preference for utilization of glucose due to the upregulation of the carbon catabolite repression system that repress the fungus to utilize other types of carbohydrates in the presence of glucose during liquid growth (Mohamed et al. 2021: Mohamed RA, Ren K, Mou YN, Ying SH, Feng MG. Genome-Wide Insight into Profound Effect of Carbon Catabolite Repressor (Cre1) on the Insect-Pathogenic Lifecycle of *Beauveria bassiana*. J Fungi (Basel). 2021 Oct 23;7(11):895. doi: 10.3390/jof7110895), and in that case, the other carbon sources

present in these complex organic nitrogen sources are usually complex and utilized after the fungus consumes the available glucose in the medium completely.

Taken altogether, we contend that the complex organic nitrogen sources by themselves were the main factor influencing the phenotypic traits in *B. bassiana* blastospores observed in the present study. Our next goal is to continue this study using ‘omics’ approaches to gain more insights into the molecular and biochemical mechanisms of these nitrogen sources influencing the carbon and nitrogen metabolisms in *B. bassiana* and search for key genes regulating these phenotypic responses. The context stated here was included in the Discussion section of the revised version of our manuscript to clarify this important issue when using complex organic nitrogen sources for fungal mass production.

Reviewer #1:

In this manuscript, authors provided experimental data to unveil exogenous nitrogen source affects fungal blastospores production, endogenous C:N ratio, virulence, shelf life, etc. Different cellular C:N ratio is an interesting view to explore the effect of nitrogen on fungal traits. However, the major concern is, this work seems to be a summary of phenotypical tests, without any solid evidence or clues investigate how the nitrogen supplements affects the microbial traits. On the other hand, cottonseed flour has already been proven to be a good nitrogen for *Beauveria bassiana*'s growth in previous publication, so it's not an innovative finding. In my point of view, is not qualify enough to publish unless authors add results and discussion associate with the mechanism research. In addition, The C:N ration displayed in Table 1 doesn't match the data shown in figure. In Table 1, the C:N ratio of "cottonseed flour" is same as "corn gluten meal" with 5:1. It should be incorrect.

Response: We appreciate the relevant and critical comments raised by the reviewer, but we disagree that our work is not qualified enough to be published without the addition of new results associated with the mechanism research. The mechanism research is, per se, another entire project of a PhD or Postdoc upon the availability of new funding that we intend to investigate in the near future based on our findings reported in the present work. The phenotypical characterization of blastospores produced with different complex organic nitrogen sources is novel and reveals the impact of these nutritional compounds on fungal fitness and virulence, which is the first step before diving into the underlying molecular or biochemical mechanisms of the fungal cell. It is really hard to point out which specific protein or amino acid was more critical or relevant to the fungal traits, since we used complex organic nitrogen compounds based on proteins and other oligopeptides, and some of them containing free amino acids, as well as composed of other nutrients such as carbohydrates, vitamins, and minerals. It is impossible to find out a specific protein or amino acid affecting fungal growth and metabolism when using complex organic compounds. This specific approach would be possible if knowing the nutritional composition of each nitrogen source and then testing a defined medium using their amino acids to make a synthetic defined medium. However, again, these complex organic nitrogen sources contain varied concentrations of minerals and vitamins, which can also contribute to fungal growth and changes in fungal fitness, virulence, and shelf life. These nitrogen sources do influence fungal biological traits such as fitness, cellular (endogenous) C:N ratios, virulence, and other important parameters required by industry, such as growth kinetics in

the bioreactor, desiccation tolerance and shelf life. In addition, we also found that these organic nitrogen sources affect the endogenous C:N ratio of blastospores, and this topic was analyzed and discussed in the present work. To date, there is no such comprehensive study investigating the effects of various complex organic nitrogen sources in fungal nutrition, fitness, virulence, desiccation tolerance, and shelf life altogether and making correlations between them. We performed robust statistical analyses with a set of solid and consistent data sets generated with repeated experiments performed for each phenotypical trait of interest. Furthermore, this work sets the foundations for future studies addressing molecular mechanisms involved in the nitrogen and carbon metabolism regulation in the fungal blastospore which genes are up and downregulated. These genes, therefore, can be manipulated to generate mutants for investigation of gene functions during blastospore formation.

In the case of cottonseed flour (with a cost of 0.89 USD per L of our medium), we confirmed that it is an excellent and inexpensive source (see Table 1 with the costs per kg of each nitrogen source) to use in *B. bassiana* blastospore fermentation as indicated in previous studies of our group. However, cottonseed flour has not been tested for this fungal strain or compared with other complex organic nitrogen sources. We also identified other nitrogen sources that could replace the cottonseed flour, with very similar results of phenotypic traits. It is important to show that there are other nitrogen sources besides cottonseed flour that might be useful for fungal fermentation by the industry. Therefore, we found additional nitrogen sources comparable to cottonseed flour that may be explored by industry for other fungal strains, as each strain will require a new study to optimize the best carbon and nitrogen sources for optimal media formulation.

Regarding Table 1, the C:N ratio of “cottonseed flour” is 4.97 while “corn gluten meal” is 4.67, according to our chemical analysis for C and N contents. These ratios are very close. We corrected this issue in Table 1 and expressed C:N ratios with double digits for all nitrogen sources tested.

Reviewer #2:

This was a well written and interesting manuscript on the effect of different complex nitrogen sources on blastospore production of *B. bassiana*, viability under different conditions and virulence. The data presented and the research approach were excellent and I found it a very interesting and useful study. I have a few points though. The manuscript is written as if nitrogen was the sole driver of the differences, although in several parts it is acknowledged that the sources are complex and contain other compounds that will affect the blastospores (e.g. line 644 onwards). Given the C:N ratios of the sources, it really isn't just nitrogen. This does not detract from the results, but the emphasis of the paper is wrong.

Response: The reviewer made a good point regarding the fact that complex organic nitrogen sources do have nutrients rather than nitrogen, but also carbohydrates, vitamins, and minerals. It is, however, impossible to point out which other nutrients present in these organic nitrogen sources could play a role in the phenotypic traits of *B. bassiana* blastospores, as we do not know

the nutritional composition of all these nitrogen sources studied here. To minimize any issue of deficient mineral and vitamin supplementation, we used a basal medium containing the basic and essential minerals and vitamins to equalize all culture media supplemented with different nitrogen sources, as these nitrogen compounds have different nutritional compositions.

On another subject, although C:N ratio of these nitrogen sources is variable, such differences are small between sources, thus playing a minor role in the phenotypic results obtained with blastospores. On the other hand, we did find that these complex nitrogen sources affected the cellular endogenous C:N ratio of blastospores after fermentation, which is one of the topics discussed in our manuscript and correlated with the fungal phenotypic variables. Endogenous C:N ratio of blastospores may be used along with other endogenous cellular nutritional reserves as a proxy to determine the quality and fitness of fungal blastospores. Please see lines 622-624, where we stated the following in the discussion section: “These nitrogen compounds also played a crucial role in altering the endogenous C:N ratio in blastospores, which could explain the notable differences in their quality, fitness, and shelf life during storage”.

Our title also states “complex nitrogen sources” which implies that there are other nutrients within these compounds. We also emphasized in the manuscript title that nitrogen sources affected the quality of blastospores in terms of cellular endogenous C:N ratio.

The authors talk about the contribution of media to the cost of production, but provide no cost comparisons of their own different sources. The medium composition in the paper is quite expensive regardless of the nitrogen source in my opinion.

Response: We disagree that our medium composition is expensive. We provided the average cost of each nitrogen source tested, as the additional components in the medium (glucose, vitamins, and minerals) were all the same and constant. The only nutrient that really impacted the medium cost was the source of nitrogen tested. We provided the costs per L of medium at 3% of nitrogen source in Table 1.

Is 4°C an appropriate storage temperature when companies are looking for shelf life at room temperature?

Response: Yes, it is the standard protocol to assess the shelf life of blastospores. In future studies, we intend to evaluate the shelf life of dried blastospores produced with the most promising nitrogen sources during storage under non-refrigerated (room temperature) conditions.

There was no variation in the % in the media of the nitrogen source- has this been looked at previously?

Response: Yes, we did look at what would be the optimal % of nitrogen source in liquid media to produce *B. bassiana* and *Cordyceps javanica* or *Cordyceps fumosorosea* blastospores and the range is between 1.5 and 3% total N (please, see references below). Of course, changing the

fungal strain or species, the nitrogen content in the medium should be optimized, as different strains and species of fungi may have different nutritional requirements.

Mascarin GM, Jackson MA, Kobori NN. 2018. Nitrogen sources affect productivity, desiccation tolerance and storage stability of *Beauveria bassiana* blastospores. *Journal of Applied Microbiology*, 124:810–820. Doi: 10.1111/jam.13694

Cliquet S, Jackson MA. 2005. Impact of carbon and nitrogen on the quality, yield and composition of blastospores of the bioinsecticidal fungus *Paecilomyces fumosoroseus*. *J Ind Microbiol Biotechnol* 32:204–210

Specific comments:

Line 103 in vitro in italics

Response: Thank you for this comment. The change is made as suggested.

Line 121-122- which reference covers which aspect? Especially which papers report blastospore infectivity or toxicity?

Response: The changes are made as suggested (now lines 126-129). As some references were added or moved, the correspondent numbers in the text and “Reference” list have also been changed.

Line 179-180 Inoculated with spores to establish a concentration of blastospores? Do you mean final concentration of spores?

Response: Thanks for the opportunity to clarify this question. We made a mistake when we referred to “a final concentration of 1×10^6 blastospores mL⁻¹”. We meant 1×10^6 conidia/mL. The sentence is now revised and re-written (see line 186).

Line 208 what pore size of the cheesecloth?

Response: We used a ~30 µm pore size cheesecloth. The information is now added (see line 214-215).

Line 275 mycelia

Response: The word is now corrected.

Line 279 freeze-dried?

Response: We apologize for this typo. The word is now corrected.

Line 310- Not clear what the volume of 2×10^{10} blastospores was? I assume this was straight from the fermenter. What was the loss through filtering at this step?

Response: The biomass was harvested from the bioreactor after 72 h of fermentation, quantified, and mixed with diatomaceous earth (DE) in the ratio of 1g DE for each 2×10^{10} blastospores. This is a well-known methodology that allows harvesting blastospores with a minimum loss, as described by Jackson and Jaronski (2012) (reference #9 in the manuscript). The sentence is now rephrased (see lines 318-320).

Line 321 does these air-dried blastospores mean with or without diatomaceous earth?

Response: In this paper, when we say “air-dried blastospores” we mean blastospores mixed with diatomaceous earth, vacuum filtered, and dried.

Line 375 what does "marked" mean here?

Response: We meant “pronounced” or “remarkable”, thus in the sentence we rephrased to “...yeast had a pronounced blastospore production...” (line 385).

Line 497 what is the pronounced impact? Next sentence says similar?

Response: In the first sentence we meant that higher blastospore yields were achieved when higher glucose consumption was observed (indirectly assessed through glucose levels in the spent medium). We rephrased sentences to improve understanding (lines 508-511).

Line 636 I wouldn't use the word "remarkable" as its not very specific nor accurate. Just higher than would be better

Response: Thanks for your comment. The change is made as suggested (lines 668).

Line 644 It is not clear from the results that nitrogen was what was optimised, given the small differences in C:N ratio between some of the sources

Response: Yes, the C:N ratio between some of the nitrogen sources is quite similar or a little different; that’s why the C:N ratio of these nitrogen compounds might not be a good factor in explaining the results. Instead of using the C:N ratio of nitrogen sources, we analyzed and reported the final C:N ratio from the cell composition of dried blastospores after produced with these various nitrogen sources and then tried to make a correlation of this variable with the other phenotypic traits of blastospores. In addition, we agree that this study was not media optimization and thus we changed the title of the manuscript by removing the word “optimizing”, as suggested by the reviewer.

Line 670 There is no comparisons mentioned in the text between conidia and blastospores, so hard to determine the comparative value of producing one over the other. Beyond the scope of this paper though.

Response: I think there is a misunderstanding about our argument at now line 713, as we are not comparing production costs between blastospores and aerial conidia. Rather, we argued that in spite of blastospores being deemed more sensitive cells than aerial conidia to abiotic (environmental) stresses, manipulating the nutritional composition of media with nitrogen

sources can lead to more robust and resilient blastospores to heat and UV-B stresses (see lines 713-717).

Line 685 The 72% of the cost as media components is very unlikely, given the cost of labour in most countries and equipment depreciation etc. A table of comparative cost of each nitrogen source would have been useful if this is your argument.

Response: Thank you for this comment. The information is now rephrased. Also, information regarding the costs of the nitrogen sources used in our study is now added to Table 1.

Line 693 complex nitrogen source

Response: The change is made as suggested.

Line 698 yields are good, not exceptional.

Response: For 3 days of fermentation, yields higher than 1×10^9 blastospores/mL can be already considered excellent when compared with previous studies of our group and elsewhere.

Line 768 space needed between types and and

Response: The change is made as suggested.

Lines 796, 916 Italics for species

Response: The change is made as suggested.

Re: Spectrum04040-23R1 (Complex Nitrogen Sources from Agro-Industrial Byproducts: Impact on Production, Multi-Stress Tolerance, Virulence, and Quality of *Beauveria bassiana* blastospores)

Dear Dr. Gabriel Moura Mascarin:

Thank you for resubmitting your manuscript to Microbiology Spectrum. As you will see your paper is very close to acceptance. Please modify the manuscript along the lines recommended by the reviewer at the end of this email. As these revisions are quite minor, I expect that you should be able to turn in the revised paper in less than 14 days.

Revision Guidelines

Sincerely,
Lea Atanasova
Editor
Microbiology Spectrum

Reviewer #2 (Comments for the Author):

This manuscript now reads well and the queries of the reviewers have been addressed, or at least discussed.

Line 395 Unaffected isn't a common English word. "had no effect" would be better. Or "are unaffected"
line 471 shown there were no significant...
line 549 gold standard

March 28, 2024

Microbiology Spectrum,
Editorial Office, ASM Journals

Manuscript #: Spectrum04040-23

Response to reviewers

Reviewer #2:

This manuscript now reads well and the queries of the reviewers have been addressed, or at least discussed.

Specific comments:

Line 395 Unaffected isn't a common English word. "had no effect" would be better. Or "are unaffected"

Response: Thanks for your comment. The change is made as suggested.

line 471 shown there were no significant...

Response: The change is made as suggested.

line 549 gold standard

Response: The word is now corrected.

Re: Spectrum04040-23R2 (Complex Nitrogen Sources from Agro-Industrial Byproducts: Impact on Production, Multi-Stress Tolerance, Virulence, and Quality of *Beauveria bassiana* blastospores)

Dear Dr. Gabriel Moura Mascarin:

I am happy to inform you that your manuscript has been now accepted, and I am forwarding it to the ASM production staff for publication. Your paper will first be checked to make sure all elements meet the technical requirements. ASM staff will contact you if anything needs to be revised before copyediting and production can begin. Otherwise, you will be notified when your proofs are ready to be viewed.

Sincerely,
Lea Atanasova
Editor
Microbiology Spectrum